# Ecology and Evolution in the RNA World Dynamics and Stability of Prebiotic Replicator Systems

**DOI:** 10.3390/life7040048

**Published:** 2017-11-27

**Authors:** András Szilágyi, István Zachar, István Scheuring, Ádám Kun, Balázs Könnyű, Tamás Czárán

**Affiliations:** 1Evolutionary Systems Research Group, MTA, Centre for Ecological Research, Hungarian Academy of Sciences, Klebelsberg Kuno u. 3, 8237 Tihany, Hungary; and.szilagyi@gmail.com (A.S.); istvan.zachar80@gmail.com (I.Z.); istvan.scheuring@gmail.com (I.S.); kunadam@caesar.elte.hu (Á.K.); 2Center for the Conceptual Foundations of Science, Parmenides Foundation, Kirchplatz 1, 82049 Pullach/Munich, Germany; 3MTA-ELTE Theoretical Biology and Evolutionary Ecology Research Group, Department of Plant Systematics, Ecology and Theoretical Biology, Eötvös Loránd University, Pázmány Péter sétány. 1/c, 1117 Budapest, Hungary; 4Department of Plant Systematics, Ecology and Theoretical Biology, Eötvös Loránd University, Pázmány Péter sétány. 1/c, 1117 Budapest, Hungary; konnyu@caesar.elte.hu; 5Biocomplexity Group, Niels Bohr Institute, Copenhagen University, Blegdamsvej 17, 2100 Copenhagen, Denmark

**Keywords:** RNA-world, ribozymes, coexistence, ecological stability, evolutionary stability, template replication, modelling the origin of life, evolvability

## Abstract

As of today, the most credible scientific paradigm pertaining to the origin of life on Earth is undoubtedly the RNA World scenario. It is built on the assumption that catalytically active replicators (most probably RNA-like macromolecules) may have been responsible for booting up life almost four billion years ago. The many different incarnations of nucleotide sequence (string) replicator models proposed recently are all attempts to explain on this basis how the genetic information transfer and the functional diversity of prebiotic replicator systems may have emerged, persisted and evolved into the first living cell. We have postulated three necessary conditions for an RNA World model system to be a dynamically feasible representation of prebiotic chemical evolution: (1) it must maintain and transfer a sufficient diversity of information reliably and indefinitely, (2) it must be ecologically stable and (3) it must be evolutionarily stable. In this review, we discuss the best-known prebiotic scenarios and the corresponding models of string-replicator dynamics and assess them against these criteria. We suggest that the most popular of prebiotic replicator systems, the hypercycle, is probably the worst performer in almost all of these respects, whereas a few other model concepts (parabolic replicator, open chaotic flows, stochastic corrector, metabolically coupled replicator system) are promising candidates for development into coherent models that may become experimentally accessible in the future.

## 1. Introduction

Prebiotic systems are assemblages of dynamically coupled replicative entities hypothesized to have existed before biological evolution, during the chemical evolutionary phase of molecules leading to the first cells and life, about 3.5–4 billion years ago on Earth. The idea of prebiotic evolution is not limited to our planet, of course: any habitat in the universe offering suitable physical-chemical conditions for the emergence and maintenance of such replicative entities may have undergone similar evolution. The units of evolution at the prebiotic era on Earth were molecular replicators (most probably RNA molecules) and their evolution may have led to the emergence of the first chromosomes and, ultimately, to the first cells. None of the recent, highly evolved biochemical machinery controlling and regulating the replication of information (such as modern error-correction mechanisms of DNA copying) had existed then. Therefore, some serious obstacles had to be overcome on the evolutionary route leading to the first individual cells and biological evolution.

The first such problem that prebiotic systems may have faced was transgressing the information threshold, i.e., escaping Eigen’s paradox. The paradox poses the following issue: The critical amount of information within a replicator system that is sufficient to keep it running through many generations is constantly ruined by mutational loss. Lacking a reliable replication mechanism, the mutation rate was probably very high. As a consequence, the critical amount of information could not be condensed into a single, long replicator, because copying errors (mutations) would have easily eroded much of the vital information in a single step. Maintaining a sufficient diversity of different replicator species, each containing a small, more reliably replicable part of the critical information, could be the solution to the information threshold problem [1]. The combined information content of such a maintainable replicator set may have been sufficient to code for a viable system. However, several other system-dependent problems had to be solved by even the simplest prebiotic replicator system. We have defined a minimum set of system-level criteria that any prebiotic replicator set would certainly have had to meet in order to be able to maintain itself for a sufficiently long time and evolve toward higher complexity:**Ecological diversity**—maintaining the coexistence of a sufficient number of different species (replicators, sequences, genotypes, etc.) in light of the Gause-principle (see later), which poses a strict limit on the number of coexisting species based on the number of regulating factors.**Ecological stability**—maintaining dynamical stability in a given set of coexistent species against external perturbations.**Evolutionary stability**—maintaining an adequate amount of information (a critical diversity of replicator species) from generation to generation and avoiding information decay (diversity reduction) in spite of frequent mutations and the lack of error correction.

Any model intended to represent the dynamics of prebiotic systems must satisfy at least these three criteria (beyond biological plausibility and interpretability). Note that there are more criteria to be met by the replicators themselves for complex life to unfold from the prebiotic systems they constitute. For replicators to be the units of open-ended [2] evolutionary change, they have to be capable of unlimited heredity [3], self-referentialism and evolution of evolvability [4], etc. (for a summary, see [5]). In this paper, we focus our attention on the diversity and stability aspects of replicator communities as emphasized above, assuming that all other requirements are met by the constituent replicators.

We will use the term “replicator” for any kind of biological or chemical entity that is capable of replication in the broadest sense (see [6]), i.e., is multiplying, has variations that affect its reproduction/survival and is creating more of its type (with variations being heritable). Mutations will play a crucial role in the evolutionary dynamics of species of replicators, the time scale of which may or may not be substantially different from that of ecological changes, depending on the actual model.

In the following section—after a brief methodological characterization of dynamical modelling—we will clarify the concepts of diversity maintenance (coexistence), ecological stability and evolutionary stability in some detail. Then we will investigate a set of models previously introduced, along the lines of these three criteria, under a separate heading for each version (differing in spatial and/or temporal resolution) of each dynamical scenario. Our aim is to provide a comparative review of the field’s most important models. We will confine our focus on models of linear polymer replicators (string replicators) and will not survey models dealing only with higher-level (compositional) dynamics such as the GARD model [7,8,9] or the models of autocatalytic sets [10,11,12], as those models can be understood as special cases of others discussed in this paper (for critical analyses of GARD, see [13,14]).

## 2. The Three Pillars of Prebiotics

Prebiotic systems are usually investigated by dynamical models. In turn, we will discuss some of the most thoroughly studied ones. Dynamical models can be classified into different categories depending on certain aspects of the dynamics they assume. The two most important such aspects are temporal and structural resolution: models may be discrete or continuous in time and they may or may not postulate spatial, group or other structure with local interactions.

Continuous time models are formalized as differential equations specifying the state of the system at *t* + *dt*, based on the state at *t*, where *dt* is an infinitesimally short time period. Temporally discrete systems are difference equations or update rules that define the state of the system at time *t* + 1 as a function of its state at *t*. Spatially structured models can be treated in continuous space by partial differential equations (PDEs) or in discrete space, as cellular automata (on different types of grids or lattices). Note that the analysis of PDE models requires numerical methods in almost any case, just as the lattice models they approximate. Since the corresponding lattice model is usually much easier to handle and it can approximate continuous time by sequential random updating rules, PDE models play a minor role in studies of replicator dynamics.

### 2.1. Maintaining Diversity

Ideal populations of replicators not limited by external factors exhibit exponential growth. For any biological entity (or replicator), the size of offspring in the population is proportional to the actual number of reproducing entities in the population (or to the whole population if everyone reproduces). In models of population dynamics, the factor of proportionality is the Malthusian growth rate, characteristic of the species, denoted by *r* (*r* > 0). Thus, the continuous-time dynamics of a population that grows without any internal or external limitation is the following:(1)$$\dot{x}\left(t\right)=rx\left(t\right),$$ where x(t) is the amount (or concentration) of a replicator species at time *t* and $\dot{x}\left(t\right)$ is the time derivative. The solution of this differential equation is the well-known exponential growth formula $x\left(t\right)=x\left(0\right)e^{rt}$, defining the actual population size at any time t. Exponential growth would increase population size beyond all limits, whereas the growth of every natural population slows down and ultimately stops growing due to the exhaustion of the limiting resource (food, space, etc.). Such regulating factors are extremely important in the coexistence of different replicator (or biological) species (see later).

The interesting dynamics arose when multiple different species are competing for the same resource. Assume two replicators with Malthusian growth rates $r_{1}$ and $r_{2}$. The ratio of their numbers at time *t* is (2)$$\frac{x_{1}\left(t\right)}{x_{2}\left(t\right)}=\frac{x_{1}\left(0\right)}{x_{2}\left(0\right)}e^{\left(r_{1}-r_{2}\right)t}$$of which the limit at t=∞ is:(3)$$\underset{t\to \infty }{\lim}\frac{x_{1}\left(t\right)}{x_{2}\left(t\right)}=\{\begin{array}{c} 0,\;\;if\;r_{1}<r_{2}\\ \infty ,\;\;if\;r_{1}>r_{2}\\ \frac{x_{1}\left(0\right)}{x_{2}\left(0\right)},\;\;if\;r_{1}=r_{2} \end{array}$$meaning that the replicator with the higher replication rate exponentially outcompetes the inferior replicator. The inferior species becomes extremely diluted in finite time, which practically means its extinction. The relative growth rate of the competitors is the difference between their Malthusian parameters. Coexistence of the two replicator populations is impossible without a mechanism that ensures the two growth rates to settle at exactly the same value. In case of any arbitrarily small difference between *r*_1_ and *r*_2_, the difference in the densities grows exponentially. This is the core problem of maintaining diversity of exponentially growing populations—and exactly the same dynamics are the indispensable basis of natural selection. It is the requirement of both maintaining diversity and remaining selectable that makes the problem particularly difficult.

As mentioned before, coexistence requires *regulating factors* which mitigate exponential competitive exclusion [15] and thereby allow coexistence. Gause’s CCCC principle (“Complete Competitors Cannot Coexist”) can be rephrased using the concepts of modern ecology: the number of coexisting species cannot exceed the number of regulating factors in equilibrium. We may consider any factor a regulating factor if: (i) it affects the growth rate of a species and (ii) it is affected by the number (density) of the same (and possibly also some other) species. Any factor that is a regulating factor for at least one of the species in a given species pool increases the possible number of coexisting species and the robustness of their coexistence. Note that the identification of regulating factors is sometimes trivial (e.g., the number of limiting resources or self-inhibition) but often it is more difficult (e.g., spatial constraints, stochasticity, periodic solutions, etc.) Furthermore, the determination of coexisting species (and hence their number) may be also complicated (as is the case for replicating nucleotide sequences, where a complementary pair of strands counts as a single replicator instead of two [16]. The presence of different regulating factors increases the chance of coexistence by relaxing competition. This is obvious in case of the self-inhibition of replicators, for which the dynamics takes the following form: $\dot{x}\left(t\right)=rx^{1/2}\left(t\right)$; for details see [17] and the section about parabolic replicators in this paper.

Beyond the occurrence of additional regulating factors, the intrinsic variability of the dynamics can also act as a factor facilitating and maintaining diversity. Periodic or chaotic variation of densities in time generated by the dynamics itself (intrinsic fluctuations) can help to maintain diversity, the fluctuations themselves acting as regulating factors, see e.g., [18,19,20].

### 2.2. Ecological Stability

While regulating factors affect the growth rates and are affected by the densities, other *external factors* may also affect growth rates (mortality and fecundity) but remain unaffected themselves by the densities of replicators. Such factors are usually abiotic, such as temperature, pH or humidity. The robustness of a fixed set of coexistent replicator species (community) against changes in external factors is the key concept of ecological stability. If typical external perturbations can cause a system to collapse or a reduction in the number of coexisting species, the system is considered ecologically unstable.

Assuming that the change in external factors occurs on a time scale shorter than that of evolutionary changes (the accumulation of mutations), ecological robustness applies to a fixed set of species, even if ecological and evolutionary time scales overlap in prebiotics (discussed in turn), due to the large mutation rates involved. Note that the variation of external factors is not necessarily detrimental to an established community; environmental variation can also act as a potential diversity-maintaining factor, as it is well known in ecology and discussed in the previous section.

### 2.3. Evolutionary Stability

Because of the high mutation rates in prebiotic scenarios, there is no clear distinction of “ecology” and “evolution” in terms of time scale separation in the dynamics. The evolutionary stability of a system means the robustness of the resident species against any invading mutant. If there are mutants that corrupt the system or reduce the number of coexisting species (thus decreasing the sustainable amount of information), the system is considered evolutionarily unstable, for detailed analysis, see [21]. Even though this aspect is often disregarded, any candidate model of a prebiotic system must meet the evolutionary stability criterion, otherwise it is seriously underestimating the potential effects of mutations. Evolutionary stability against deleterious mutants is at least as important as ecological stability, precisely because of the overlapping time scales.

An indispensable aspect of “forward” evolutionary stability is evolvability: the propensity of the system to adopt new replicator species (possibly originating as mutants of existing ones or supplied from outside) if they are of any advantage in terms of the collective fitness of the system. (See the corresponding group selection arguments later, in Section 3.2.3 and Section 3.2.4). All the models discussed in this paper will be scrutinized also in this respect, by assessing the probability of the given system to produce and incorporate beneficial mutants.

## 3. Models of Prebiotic Systems

In this section, we will scrutinize some of the most important and mainstream dynamical models of prebiotics with respect to their ecological and evolutionary stability properties. We aim to analyse the explanatory power and applicability of these models in the context of the three criteria explicated in the Introduction above. Specifically, our analysis includes the following models:The Quasispecies model (QS) [1,22]The Hypercycle (HC), spatial hypercycle (SHC) [23,24,25,26], compartmentalized hypercycle (CHC) [27] modelsThe Parabolic replicators (PR) model [17]The Stochastic Corrector model (SCM) [27,28,29,30]The Open chaotic flow (OCF) model [31,32]The Metabolically Coupled Replicator System (MCRS) model [33,34,35,36,37,38]Trait Group Models (TGM) [39,40]

Table 1 categorizes these model types on the basis of their temporal and structural resolution; Figure 1 provides a “genealogy” of the models.

### 3.1. Models Assuming no Structure

Models without spatial or compartmental structure can be easy to deal with, as there is no need to account for the corresponding spatial aspects of the dynamics, so that local differences in concentrations/amounts, limited ranges of interactions and localized physical processes (droplet formation, diffusion, bonding to surface, vesicle division, etc.) can be drastically simplified or even omitted. Mean-field simulations are usually easy to approximate analytically. On the other hand, the lack of any structure means that these models have a limited ability to maintain diversity.

#### 3.1.1. Hypercycle (HC)

The hypercycle was proposed by Eigen and Schuster [22,41,42,43] as a solution to the error threshold [1], a severe limit to the information content of primordial biological sequences. The replication of information-carrying macromolecules is prone to error [44] and the error rate (mutation rate) was higher at the origin of life [45], due to the lack of effective and high fidelity replicase enzymes and proofreading mechanisms. A functional sequence is replicated but some of its progeny will be of a different—most probably non-functional—type due to mutations. The following equations describe a system of a replicating functional, wild-type sequence (its concentration denoted by *x*_w_) and all of its possible mutants lumped together (their total concentration denoted by *x*_m_):(4)$${\dot{x}}_{W}=x_{w}\left[QA_{W}-\Phi \right]$$(5)$${\dot{x}}_{m}=x_{m}\left[A_{m}-\Phi \right]+\left(1-Q\right)A_{W}x_{W}$$where *Q* is the probability of faithful replication of a sequence; *A_W_* and *A_m_* are the replication rates (Malthusian growth rates) of the wild-type sequence and its mutants, respectively; and Φ is the outflow term to keep the total concentration constant. It is evident that in such a system the wild type will go extinct if *A_m_* > *A_W_*. Coexistence, i.e., the survival of the wild type is only possible if *QA_W_* > *A_m_*. Mutational rates are often expressed in units of mutation/nucleotide/replication (*μ*) instead of replication fidelity. Given a sequence of length *L*, the fidelity of replication is *Q* = (1 − *µ*)*^L^* ≅ *e^−Lµ^*. We can then arrive at the inequality of the error threshold setting an upper limit to reliably replicable sequence lengths:(6)$$L<\frac{\ln\left(A_{W}/A_{m}\right)}{\mu }$$

Assuming that the per nucleotide mutation rate is 1% [45] (which is a realistic assumption for replications unaided by efficient enzymes) and that the wild type has better replication rate than any mutant at least conforming to the ln(*A_W_*/*A_m_*) = 1 relationship, we find that a wild type sequence of length 100 but not more, can be stably maintained. Note that since the threshold expression (Equation (6)) is proportional to the *logarithmic* ratio of the functional and the non-functional replication rates, increasing the replication rate of the wild-type does not increase the length of the maintainable sequence too much. This result yields Eigen’s Paradox [46]: there is no accurate replicase without a large genome and there could be no large genome without an accurate replicase. Thus, the information that can be reliably replicated is less than the information necessary to code for the replicating machinery. This is a key dynamical problem to which the early evolution of life had to find a solution [47,48,49].

The hypercycle was devised to overcome Eigen’s Paradox. If a single sequence cannot maintain enough information, then the necessary amount of information needs to be replicated in several sequences. Information stored in short sequences can be replicated, whereas the same amount of information in a single sequence may be far above the error threshold assuming the same mutation rate. However, the different sequences will inevitably compete with each other and given the limited number of resources (monomers) and the lack of other regulatory constraints, only one (or as many as there are different resources) of the sequences will survive. Thus, a mechanism is required to establish cooperation among the sequences so that none of them outcompete the others. In the hypercycle, each replicator (sequence) catalyses the replication of another sequence in the set. Each replicator catalyses the replication of only one other replicator and receives catalytic aid from only one other replicator, the interaction thus occurring in a circular topology. For example, in a three-membered hypercycle R1 catalyses the replication of R2; R2 catalyses the replication of R3; and R3 catalyses the replication of R1 and closes the hypercycle (see Figure 2).

Formally, the concentration of a replicator *i* in an *n*-membered hypercycle can be written as(7)$${\dot{x}}_{i}=x_{i}\left[Q\left(A_{i}+K_{i}x_{i-1}\right)-\Phi \right]$$where *K* is the catalytic aid received from the previous member in the hypercycle, *i* = 1 … *n*, (*x*_0_ ≡ *x_n_*); all other symbols are as above.

We need to stress here that members of the hypercycle catalyse the formation of the next member but they themselves are not converted to the next member (i.e., reactions are second order of the form R1 + R2 → R1 + 2R2). There is a lingering misconception in the literature [50] which results in the cyclic (first order) production of certain molecules being called a hypercycle, which it is not.

How efficient is a hypercycle in integrating information, i.e., how many functional sequences could coexist in it? The higher the number of coexistent sequences, the more information the system maintains. If for all *i*, *A_i_* = 0, i.e., the replicators cannot replicate on their own, only with the help of another catalyst, then the system is fully cooperative and all members coexist [51,52,53,54,55,56]. This is the homogeneous hypercycle. Assuming that all catalytic rates are the same, the dynamics leads to a stable fixed point for two-, three- and four-membered hypercycles [51,52]. Furthermore, if there are five or more members in the hypercycle, then the system approaches a stable limit cycle. Theoretically, any number of sequences can coexist but with high numbers of members some replicator concentrations may decrease to very low values during oscillations and with any one of the members lost the whole system collapses. Therefore, for *n* > 4 the system is unstable.

In the inhomogeneous hypercycle (*A_i_* > 0) the members are also in competition and if the *A_i_* values are too large compared to the *K_i_* values, then one or more of the sequences can be lost [54]. Again, hypercycles with *n* ≤ 3 members converge to a stable fixed point [51] and ones with five or more members exhibit oscillatory behaviour (stable limit cycles [57,58]. Hypercycles of six or more members can be unstable [59]. Stability is further affected by differences in the catalytic aid members give to each other [60]. In conclusion, we may say that the hypercycle can show rich dynamics [61,62], although its ability to maintain the coexistence of even a moderate number of different replicators is limited.

So far, we have not considered the quasispecies, i.e., the cloud of mutants generated around the wild-type sequence in a hypercyclic system. We can lump all mutants together and follow their concentration in a way similar to that of Equation (5):(8)$${\dot{x}}_{m}=x_{m}\left[A_{m}-\Phi \right]+\left(1-Q\right)\left[\overset{n}{\underset{i=1}{\sum }}\left(A_{i}x_{i}+K_{i}x_{i}x_{i-1}\right)\right]$$

Analysing the dynamics of hypercycles and the mutants of the master sequences uncovered a new threshold [63,64]. A replication fidelity lower than the error threshold does not allow for the maintenance of a single long molecule but shorter sequences organized into a hypercycle can coexist with their mutants. There is a lower critical copying fidelity below which even the hypercyclic organization collapses, because the mutants overwhelm the system. Yet there is a range of copying fidelity which does not allow a single long molecule to coexist with its mutants but the same amount of information arranged in a hypercycle can be maintained.

Silvestre and Fontanari [65] have cast some further doubt on the information integration capability of hypercycles. While they were able to show that even long hypercycles with *n* = 12 can be maintained, the copying fidelity puts an upper limit on the number of sequences (*_n_*) that can coexist. They find that if all *A_i_* are the same (*A*) then (9)$$n<\frac{Q^{2}}{4A\left(1-Q\right)}$$

Thus—they argue—chopping up the information into many smaller bits does not help. On the other hand, differences in replication and catalytic rates can ensure that information in many pieces can be maintained whereas a long chromosome cannot [66].

The hypercycle as an organization is capable of information integration. The question now is whether it is capable of evolution toward increased information content? Once a hypercycle is established, it is difficult to replace it with another hypercycle [41]. The hypercycle as a whole system exhibits hyperbolic growth and entities initially having a higher population size have an advantage in such a growth regime [67]. Even if we start from the same concentration, no coexistence of competing hypercycles is possible [68].

Catalytic species sometimes also inhibit some reactions. The hypercycle was also studied considering inhibitory/suppressive interactions. If there is strong suppression, then even-membered hypercycles cannot maintain all their species, whereas odd-membered hypercycles can. But even-membered cycles outcompete odd-membered cycles and thus the hypercycle generally breaks down under strong suppression [69].

As a consequence, while half a dozen sequences can possibly coexist, the system cannot evolve to incorporate more members. The evolutionary potential of the hypercycle is thus severely limited.

Niesert and co-workers [70] and Maynard Smith [71] pointed out a series of even more severe problems of the hypercycle that arise if mutations are allowed. There are two kinds of mutation that can destroy the system. One mutation turns a regular member to a selfish parasite, a sequence that accepts the catalytic aid given by a member but does not reciprocate (does not help the next member). If this parasite receives strong enough catalysis, then it can spread and channel away catalytic aid, leading to the collapse of the hypercycle (see Figure 3, left panel). A second class of mutation can alter the specificity of aid given to other members of the hypercycle. If a new mutant arises that helps the replication of a member of the hypercycle other than the next one in the cycle, then a shortcut forms (see Figure 3, right panel). Such a shortcut parasite reduces the hypercycle to one that consists of fewer members than the original (i.e., reduces overall diversity). A shorter hypercycle having shared members with a longer hypercycle can spread in expense of the longer one. This represents evolution to shorter and shorter hypercycles. Information is lost with each loss of a member.

So far, we have assumed that the replication rates of the wild-type sequences are higher than the replication rates of the mutants. If there is a mutant with a higher replication rate, then it could outcompete its slower master sequence. Functional sequences are usually long and their shorter mutants replicate faster, as demonstrated in Spiegelman’s experiment [72,73], in which the Q*β* phage genome was replicated without selection for function. The functional sequence was lost by the fourth serial transfer. The sequence population was evolved to be mere fifth of the length of the original sequence but it was replicated 15 times faster [72]. That is, mutations allowing faster replication for the quasispecies are all potentially deleterious to the “naked” (non-spatial) hypercycle. For a recent review on the hypercycle, see [74].

#### 3.1.2. Parabolic Replicators (PR)

The first problem that a prebiotic replicator community has to solve if it is to start up life is to avoid the competitive exclusion of its constituent replicators, i.e., to maintain a critical diversity of replicator species even in the face of the shortage of resources (for string replicators: nucleotides) that at some point inevitably occurs for any replicator population capable of exponential increase. The problem seems even more difficult to solve given the lack of other conceivable regulating factors (mainly due to the simple chemical kinetics of prebiotic systems). As we have shown in the previous section the hypercycle, the first solution attempt to the coexistence problem, may fail to be a solution for more than a few reasons. In this section, we present another simple chemical kinetics of template-based replication for a special case in which Darwinian selection does not occur and the system ends up in a “coexistence of everyone” regime. Note that in the “standard”, resource-regulated dynamics of template replication with a detailed chemical kinetics (per base elongation of sequences) a complementary pair of sequences counts as a single replicator, not the solitary strands. Consequently, four different nucleotides can maintain the coexistence of four complementary *pairs* of strands [16]. This poses a strict limit on the diversity of the coexistent replicator community in the lack of other regulating factors.

The simple kinetics of template-based replication is the following. Assume that replicator A reacts with resource R at rate *K* and produces another replicator A that remains associated with the original (AA) and there is an association-dissociation process between double and single strands (with rates *k* and *k*′, respectively):$$A+R\overset{K}{\to }\mathrm{AA},\;A+A\overset{k}{\to }\mathrm{AA},\;\mathrm{AA}\overset{k^{\prime }}{\to }A+A$$

For this type of dynamics to occur the self-association of molecules is needed, which is possible e.g., in case of palindromic sequences. An important and chemically plausible restriction is the order of the rate constant values: *K* < *k*′ << *k*, i.e., association is much more probable than dissociation. Note that the result of the replication is the complex AA which is inert to replication, thus this chemical machinery has an interesting feature: it is self-regulated—the higher the concentration of A, the stronger the self-regulating effect. The speed of replication is determined by the concentration of (dissociated) A as this can act as a template for the replication. As von Kiedrowski first pointed out in 1986 [75], chemically embodied artificial replicators (modified hexanucleotides) behave according to this type of kinetics (see [76] for a detailed analysis of dynamics and [77] for an overview of artificial self-replicators.)

This type of self-replication substantially alters the dynamics of the system; replicator concentrations undergo parabolic rather than exponential growth (cf. Equation (1) and see e.g., [75]):(10)$$\dot{x}=rx^{p}$$where *x* is the total concentration of A and AA, *p* = 1/2, $r=\rho K\sqrt{\frac{k^{\prime }}{2k}}$ and ρ denotes the concentration of R.

In almost all experimentally investigated systems *p* ≈ 1/2 (*p* = 1 corresponds to “standard” exponential dynamics, the 0 < *p* < 1 interval corresponds to the parabolic growth in a broader sense). It can be easily seen (cf. [17]) that this type of dynamics maintains the coexistence of an arbitrary number of replicators. By introducing the constraint of the total replicator concentration being 1, the dynamics of *N* different types of replicators with Malthusian parameters *r_i_* takes the following form:(11)$${\dot{x}}_{i}=r_{i}x_{i}^{p}-x_{i}\overset{N}{\underset{j=1}{\sum }}r_{j}x_{j}^{p}$$

After a simple rearrangement we get(12)$${\dot{x}}_{i}=x_{i}^{p}\left(r_{i}-x_{i}^{1-p}\overset{N}{\underset{j=1}{\sum }}r_{j}x_{j}^{p}\right)<x_{i}^{p}\left(r_{i}-Nr_{max}x_{i}^{1-p}\right)$$ where *r_max_* denotes the largest Malthusian parameter and we used the following inequality: $\overset{N}{\underset{j=1}{\sum }}r_{j}x_{j}^{p}<\overset{N}{\underset{j=1}{\sum }}r_{j}<Nr_{max}.$ Obviously, any replicator has a positive growth rate if its concentration drops below a critical threshold:(13)$$x_{i}<{\left(\frac{1}{N}\frac{r_{i}}{r_{max}}\right)}^{\frac{1}{1-p}}$$thus, at least theoretically, the advantage of rarity warrants the survival of everybody, whenever the replicators are in a competitive situation. This result was extended by Varga and Szathmáry [78] showing that there is a single internal and globally stable rest point of the system of Equation (11).

It is instructive to compare the solution of the dynamics of exponential growth, (Equation (1), or *p* = 1 in Equation (10)) and parabolic growth (Equation (10) with *p* = 1/2) for two replicators. While in exponential growth the ratio of the concentrations of the two replicators is an exponential function of time (resulting in competitive exclusion), in the parabolic case the ratio is:(14)$$\frac{x_{1}\left(t\right)}{x_{2}\left(t\right)}=\frac{{\left(\sqrt{x_{1}\left(0\right)}+\;r_{1}t/2\right)}^{2}}{{\left(\sqrt{x_{1}\left(0\right)}+\;r_{2}t/2\right)}^{2}}\underset{}{\to }\underset{t\underset{}{\to }\infty }{\lim}\frac{x_{1}\left(t\right)}{x_{2}\left(t\right)}=\frac{r_{1}^{2}}{r_{2}^{2}}$$meaning that the ratio of the equilibrium concentrations depends on the ratio of the squared Malthusian parameters (this is where the name of parabolic replication comes from). Interestingly, this result is in line with Darwin’s statement on the geometrical increase of populations: “The Struggle for Existence amongst all organic beings throughout the world […] inevitably follows from their high geometrical power of increase …” [79].

Note that for a large number of replicator types (*N* >> 1) the equilibrium concentration may be very low, so that stochastic drift can drive some replicator species extinct from the community even if their growth rates are positive. Despite this effect and the chemical constraint of the self-association of replicators, the system seems to be solving the problem of the maintenance of a critical diversity, because it is capable of storing a large amount of information (cf. Section 3.1.1, the Eigen-model and information threshold). The beneficial ability of parabolic replication to maintain diversity is itself also its most serious fault: selection (in the Darwinian sense) is not possible in this type of system. Since better replicability does not mean competitive dominance, there is no room for evolution because selection is paralyzed. As we will see later, in this sense the parabolic replicator model is homologous to the model of replicator dynamics in chaotic flows.

The analysis can be extended by treating the dynamics of single and double strands separately. In this case the selection-free regime exists only above a critical total concentration [17,75]. The explanation is straightforward: at low concentrations, single strands do not frequently pair up to form double strands; thus, self-inhibition is weaker than cross-inhibition. The selection-free regime switches to selective upon assuming the (naturally present) exponential decay of single-strands [80]. Exponential decay is the most conservative assumption for decay of atoms and molecules including replicators, with the number of decaying replicators assumed to be proportional to the number of replicators present. The behaviour of the system may change if both single- and double-strand forms can exponentially decay (even if the decay rate of the double-strand is much smaller than that of the single-strands), as it is shown in [81] for two competing replicators. In this case the outcome of competition depends on the parameters, mainly on the influx of the resource and the decay rates of single- and double-strands. At high levels of resource influx, the replicator concentration is high and thus parabolic replication and coexistence remain possible, whereas below a critical level of influx, selection sets in and the superior replicator outcompetes all the others—the influx of nutrients acts as the control parameter of selectivity.

In an attempt to extend the investigations beyond the spatially homogenous case described by Equation (10), template-directed replication was assumed to occur on a surface [82]. Double-strands bind to the surface stronger than single-strands, which in terms of decay corresponds to the assumption that single-strands have a higher decay rate. A semi-analytic investigation of the corresponding model shows that two parabolic replicators competing for their building blocks on a mineral surface are subjects of Darwinian selection under a wide range of parameter values. Differential adherence to the surface guarantees different decay rates, while the different influx of nutrients (the control parameter of different selective regimes) is due to different rates of resource supply.

Product inhibition leads to parabolic replication in non-enzymatic (artificial) replicator systems, resulting in parabolic amplification that switches off selection, consequently it cannot be the mechanism of evolutionary dynamics. A few investigations have revealed that Darwinian selection can still operate under rather specific circumstances (separate dynamics for single and double strands, exponential decomposition of strands and surface-bound template-based replication). Yet, at its present state parabolic replication seems to be of limited relevance in prebiotic evolutionary research, precisely because of its narrow scope for evolvability.

### 3.2 Models Assuming Structure

These models either assume a strict spatial order (usually on a 2D lattice) or a vesicular grouping of replicators into compartments. Either way, local interactions dominate the system, often coupled with multiple levels of dynamics and/or selection.

#### 3.2.1. Spatial Hypercycle (SHC) and Compartmentalized Hypercycle (CHC)

There could be a way out of the evolutionary problem for the hypercycle, especially from the problem posed by fast replicating mutants. If the hypercycle is localized onto a surface or into compartments, then higher level selection can weed out the parasites.

Boerlijst and Hogeweg were the first to analyse a spatial version of the hypercycle [25,26]. The spatial version of the hypercycle alleviates the problem of the stability of large systems (consisting of more than four members), thus solving one of the problems of the non-spatial version. Furthermore, the spatial spiral patterns formed conveys some resistance to parasites. A pure parasite which appears after the formation of the spirals is ousted to the outer edges of the spiral, where it decays. Parasites can kill a spiral if they are introduced exactly to the middle of the spiral. Then neighbouring spirals take over the space and thus the parasite is destroyed or lingers in a kind of “cyst.” Inhibitory effects [83] and a gradient in the decay rate of molecules [84] can further fortify the spirals against parasites. The partial differential equation model of the same system exhibit less spiral formation and it is prone to parasites that kill the spirals [23,24].

Spatial arrangement and compartmentalization [27] seems to solve the problem of a fast replicating parasite. Shortcut mutants, however, still outcompete longer sequences [85]. The short cycles that cannot form spirals are at a selective disadvantage compared to ones that can form spirals and thus exclude parasites. So, a shortcut mutant can spread and then the system becomes prone to parasites. Evolutionarily the spatial hypercycle is as limited as the non-spatial one: once established no novel, disjunct hypercycle can invade the system [25].

Based on the above and to put it bluntly: the hypercycle does not work; it is evolutionarily unstable. This is an important and rather old result that has never been circumvented. Thus, the hypercycle cannot solve the problem of prebiotic information integration. Despite its rich literature, it is time to put this model to rest.

#### 3.2.2. Coexistence in Open Chaotic Flow (OCF)

A prebiotic habitat without spatial structure is generally considered to be a set of replicators mixed intensively in an aquatic medium. However, the mixing of liquids is rarely perfect: peculiar spatial structures often emerge because of nonlinear phenomena in hydrodynamics. Open chaotic flows—one specific realization of the huge branch of complex hydrodynamical phenomena—are particularly interesting from our point of view. A flow is considered to be open if there is a continuous flux into and out of the observed region of the fluid medium and the recirculation time is much longer than the life time of the advected particles [86]. If the flow is time dependent but non-turbulent, then advected particles (replicators, in our case) move chaotically through this observed region following complex trajectories. The whole branch of possible trajectories then forms a fractal set and particles move along this fractal for a long time before they escape from the observed parcel of liquid (Figure 4).

How do populations of replicators living and multiplying in an open chaotic flow behave? The dynamics of autocatalytic replicators in two-dimensional open chaotic flows has been derived by [87] as (15)$$\dot{x}=-\kappa x+\nu x^{-\beta }$$where *x* is the number of replicators along the fractal filament and κ is the average escape rate from the observed region. The second term is the production proportional to the autocatalytic reaction rate defined above; *β* = (*D* − 1)/(2 − *D*) depends purely on the fractal dimension *D* of the filaments. Since 1 < *D* < 2 in two dimensional flows, then *β* > 0 and thus reproduction becomes more and more effective as the number of replicators decreases. This advantage of rarity is the consequence of the fractal structure itself, which therefore acts as a catalyst in generating the peculiar nonlinear population dynamics. The dynamics leads to a stable stationary equilibrium of replicators along the fractal set as *x** = (*ν/κ*)^1/(*β*+1)^ [87].

It is precisely this non-standard dynamics of replicators that leads to the advantage of local rarity, balancing the concentration differences of different replicators competing for the same limiting resource and thereby allowing for their coexistence. To formally approach this problem a two-dimensional flow is modelled around a cylinder. For a wide range of inflow velocities there is a periodic vortex detachment in the wake of the cylinder. The flow is, then, time periodic and purely deterministic but particles move chaotically in the wake of the cylinder [31]. The limiting resource flows constantly into the mixing region. Replicators were modelled as particles moving along the flow, decaying spontaneously and replicating if there are resource particles at sufficient density in the neighbourhood of a replicator. While competition for a single limiting resource leads to the survival of only the most effective replicator in a well-mixed habitat, simulations with the model have revealed that competitors coexist along the fractal set in the wake of the cylinder (Figure 5).

The results obtained by numerical simulations are reinforced by analysing the dynamics of competing replicators in open chaotic flows by mathematical means. The analysis makes use of the fact that there is only stretching and folding along the fractal set providing habitats for the competitors and thus they are arranged into more or less parallel stripes. In the simplest case with two competitors this leads to the dynamics $${\dot{x}}_{i}=-\kappa x_{i}-q\left(D-1\right)\nu x^{-\beta -1}x_{i}+qp_{i}\left(x_{1}/x_{2}\right)\nu_{i}x^{-\beta }$$where *q* is a geometric constant, *ν_i_* is the speed of the reaction front for replicator *i* (= 1,2), *p_i_* is the probability that replicator *i* is at the boundary of the habitat stripe and the resource and *ν* = *p*_1_*ν*_1_ + *p*_2_*ν*_2_ is the average reaction speed [88]. Due to dimensionality and symmetry reasons (16)$$p_{1}\left(\frac{x_{1}}{x_{2}}\right)=\frac{{\left(\frac{x_{1}}{x_{2}}\right)}^{\alpha }}{\omega +{\left(\frac{x_{1}}{x_{2}}\right)}^{\alpha }},\;p_{2}\left(\frac{x_{1}}{x_{2}}\right)=1-p_{1}\left(\frac{x_{1}}{x_{2}}\right)$$where *α* and *ω* are positive constants. The positive fixed point of this system is stable if 0 < *α* < 1. For *α* = 1, *ω* = 1 which is the case if mixing is complete; then there is no coexistence. Similarly, for *α* > 1 the initially more abundant competitor wins (over dominance) [88]. Analysis of the individual-based (IB) model of this system pointed out that *p_i_* really follows Equation (15) and, whenever coexistence is observed, the inequalities 0 < *α* < 1 hold, just as the analysis forecasts. Competition rules can be defined in different ways in the IB model. Interestingly, competitors could not coexist when some rules were applied but these rules always imply *α* > 1, as expected. (For *α* = 1 either species 1 or 2 wins the competition depending on other parameters of the model.) Thus, IB models reveal that without knowing the detailed mechanism of competition we cannot determine the dynamical behaviour of the replicators in open chaotic flows [88]. Moreover, although the analysis has been completed only for the two-species model so far, it is straightforward to extend it to many species with $\;p_{i}=\omega_{i}x_{i}^{\alpha }/\overset{m}{\underset{i=1}{\sum }}\omega_{i}x_{i}^{\alpha }$, where *m* replicators compete along the fractal set. Using the method presented in [17] it can be shown that any number of replicators coexist if 0 < *α* < 1, see Section 3.1.2. That is, the dynamics are formally equivalent to those of parabolic replication, although the subexponential term in the replication dynamics follows from imperfect mixing along the fractal set and not from the self-inhibition of replicators [89].

#### 3.2.3. Trait Group Model and Kin Selection (TGM)

The requirement of maintaining an above-minimal level of information in a replicative system can be translated to the issue of slow replicators coexisting with faster ones. For structured populations, the first models dealing with coexistence originated from social ecology. There the problem translates to whether inferior replicators (slowly replicating but altruistic in terms of providing help to the group) can survive against selfish ones. In broader terms, the question is whether a useful replicator can successfully compete with its own mutants. Answering this question requires the introduction of multiple levels of selection.

Wilson constructed his trait group model (TGM, structured deme model) [39,90] to demonstrate that group selection at the supraindividual level can lead to the coexistence of altruistic individuals with inferior fitness compared to more selfish, opportunistic ones. Altruism in this context means favouring others at the expense of the fitness of the altruist. Wilson based his group dynamics on the reproduction and dispersal of multicellular organisms. Individuals compete within local groups called “trait groups”, which are smaller than a deme but larger than a single individual. After a generation, groups disband, individuals are dispersed and mixed and ultimately form new trait groups. The model effectively separates global genetic mixing at the deme level and local ecological interactions at the trait level. Ecological dynamics in the dispersal phase are different from those in the competitive, non-dispersal phase. This distinction effectively imposes structure on the population with a new, higher level of selection in effect at the deme level. Higher level success is indirectly linked to within-group selection governed by individual growth and replication rates, as traits affect the group’s fitness. Small group sizes, low migration rates and rapid removal of compartments infected with the selfish replicator all favour group selection [91].

Wilson’s model (see Figure 6) explicitly assumes two types (two alleles of a gene) in the population. Wilson proves that “altruistic” individuals (those decreasing their own fitness in exchange for helping others) can coexist with selfish ones or increase their frequency if the variance within groups is higher than random, i.e., if the groups are not random samples of the population (see [91]). The slightest deviation from perfect genetic mixing and random reassortment could provide the required non-random distribution (for example, relatives tending to stay together) hence there is no need for compartmentation (physical separation) and Wilson’s model simplifies to any kin selection model fulfilling Hamilton’s equation [92], which requires a larger-than-zero relatedness for an altruistic trait to increase its frequency.

In Wilson’s model, individuals do not replicate within trait groups, only undergo selection, though this scenario can be replaced with replication and selection to comply with prebiotic replicators. Individuals sharing an altruistic trait correspond to cooperative replicators and individuals lacking this trait are selfish ones. Results in general are invariant for variable trait group sizes. In the general case, an all-defector population is stable against invasion by co-operators [93]. If, however, the defective replicator is in some sense dependent on the cooperative one, there is a wider scope for stable coexistence and an all-defector population may allow the invasion of co-operators. In a hypercyclic example (modelling defective interfering viruses, DIV), there are two outcomes: either the cooperative replicator wins if it helps itself more than it helps the defective one; or they will coexist and the co-operator cannot disappear [94]. Since a defector can only replicate by coupling with a co-operator (an assumption specific to the DIV model), any group consisting only of defectors perishes, ultimately increasing the overall frequency of co-operators in the population. Hence the all-defector group is evolutionarily unstable and any stochastic process generating co-operators (like mutation) would lead to their successful invasion, regardless of variable trait group sizes.

The trait group model directly relates to other models of the field. Increasing the diffusion rate in the MCRS (see Section 3.2.5) leads to the TGM with the replicative phase taking place locally (due to surface-binding) but genetic mixing is intense (due to diffusion; [33]). If compartments divide instead of intense global mixing (and replicators independently replicate within the compartments), then the SCM emerges (see Section 3.2.4).

The problem with the TGM is precisely what Maynard Smith recognized: a trait group, due to global mixing, does not form a true unit of selection but simply realizes kin selection (for example, by locally reproducing organisms forming a kin). To tap the advantage of true group selection, one must maintain group structure *continuously*, so that selection at the group level can act against groups of inferior compositions. This is what the stochastic corrector model realizes.

#### 3.2.4. Stochastic Corrector Model (SCM)

The SCM was designed [28,95] to remedy both the trait group model’s fuzzy compartmentation and lack of proper higher-level unit of selection and the hypercycle’s frailty against mutants [28,96,97].

The model implies the following steps (see Figure 7): different replicator types proliferate within vesicles (compartments, possibly “simple cells”). When the internal replicator concentration reaches a limit, the cell splits in two due to naturally emerging physical constrains within its assumed lipid boundary. If there is an optimal composition of selfish and altruistic replicators (i.e., the compartment containing them has the highest fitness at the group level), then it can be proven that this optimal composition will always be present in the equilibrium distribution of various compositions. For this to occur, the following assumptions must be met:Template replicators replicate independently within vesicles (they are not hypercyclically coupled), competing for shared resources (nucleotides, enzymes, space).Replicators contribute to a common good (e.g., metabolism) such that they affect the selection of the whole group, thus compartment fitness (group replication rate) depends on composition.Replicators are essential: a group can only replicate if both replicator types are present.The redistribution of molecules during fission is not biased by any replicator type but is random for each molecule, hence they will follow a hypergeometric distribution in the offspring.Compartment size is relatively small and fission happens before equilibrium is reached in cells.Replicator migration (or other transposon-effects) from one compartment to another is negligible.

The internal dynamics for the two replicator types $x_{1}$ and $x_{2}$ are:(17)$$\begin{array}{l} {\dot{x}}_{1}=ax_{1}{\left(x_{1}x_{2}\right)}^{\frac{1}{4}}-dx_{1}-\frac{x_{1}\left(x_{1}+x_{2}\right)}{K}\\ {\dot{x}}_{2}=bx_{2}{\left(x_{1}x_{2}\right)}^{\frac{1}{4}}-dx_{2}-\frac{x_{2}\left(x_{1}+x_{2}\right)}{K} \end{array}$$where *a* and *b* are replication rate constants; *a* > *b* ensures competition. Both types’ degradation rates are *d* and the common carrying capacity *K* ensures that the internal population of a cell cannot grow to infinity. The exponent ¼ (in fact, any exponent smaller than ½) ensures that in the limit both replicators go extinct (without group structure, of course), thus it acts as a worst-case assumption.

If vesicles were split after the internal equilibrium has settled, either both types or one of them would be extinct by the time of division according to dynamics, ultimately leading to the whole population losing information. Due to the stochastic nature of replication-degradation (“demographic stochasticity”), compartment fission and molecule reallocation, the optimal combination reappears and the distribution of probabilities for the different combinations can be calculated at group-level equilibrium [28]. Since replicators provide aspecific help to the group via a shared metabolism (like in the MCRS, see Section 3.2.5) rather than direct help to other replicators (like in the DIV model), they are not forming a hypercycle.

Szathmáry and Demeter have applied the quasispecies model to the various compartments rather than to individual molecules (assuming a finite number of compartment types; [98])—emphasizing that in this case, compartmentalized groups of replicators follow their own internal dynamics depending on their internal states. It can be shown that internal dynamics and compartment splitting lead to a dominant equilibrium quasispecies [22] in which all compartment types coexist (and thus no replicator is lost; [28,29]). This condition is always met, as the stochastic replication and reassortment of molecules ensure that each compartment composition can turn into any other [28].

In conclusion, independently replicating different replicator types functionally complement each other within a compartment. Consequently, the compartment with the optimal composition of replicators has the best fitness. Stochasticity in replication and reallocation during fission generates the necessary variation, on which natural selection at the compartment level can act [28]. The SCM effectively realizes group selection that guarantees replicator coexistence.

Zintzaras and his co-workers compared compartmentalized hypercycles (CHC) with the SCM [27]. They have found that both compartment-systems can integrate information successfully, though the SCM is able to operate under higher deleterious mutation rates and settles at a lower equilibrium mutational load than the CHC, which, however, reaches better average fitness values. The important caveat here is that compartments only contained two-membered hypercycles. Scalability (maintaining more replicator types) obviously favours the SCM, as larger hypercycles are prone to shortcut mutations. Whether fusion of compartments increases diversity and stabilizes the system in general is not clear yet, though some theoretical results indicate positive effects [99]. Hubai and Kun, under more realistic assumptions, concluded that ~100 different genes could have survived in a simple protocell [30]. In vitro realizations prove that (transient) compartmentalization is effective in maintaining a functional diversity of replicators within vesicles [40].

It is worth noting that the SCM was the first model to explicitly assume all three subsystems of cellular organization and thus life, as defined by the Chemoton model [100,101,102]: it deals with the competition of different information carrying templates, while assuming an unspecified metabolism and a boundary subsystem that encloses the composition. A stochastic implementation of the chemoton model (approximating the SCM) proves that two different competing template replicators can coexist in a protocell [103]. The SCM also realizes multilevel selection properly, hence it is modelling the result of a major evolutionary transition in which competing elements of a lower level of selection are successfully integrated at a higher level [2,104].

#### 3.2.5. Metabolically Coupled Replicator System (MCRS)

##### The Concept of a Metabolically Coupled RNA World

All the RNA-based models of prebiotic evolution are built on the assumption—which, at the time of writing, remains empirically unproven—that the template replication of the first RNA replicators was possible in the RNA World [98], even if it was slow and unreliable at the beginning. This assumption is indispensable because, for the evolutionary machinery to swing into action, populations of self-replicating entities are a necessary condition [2]. Thus far we know of no non-enzymatic RNA-replicating mechanism capable of copying reasonably long strands of RNA and it seems improbable that we can ever come up with one, so it is straightforward to postulate that RNA replication must have been enzymatic from the outset. Under the most likely prebiotic conditions, which surely did not provide the efficient peptide-based biochemical devices of recent cells, the necessary enzymatic help could not come from anywhere else but within the RNA World itself: RNA-dependent RNA polymerase ribozymes (or groups of ribozymes) that ignited prebiotic replicator evolution must have existed. However, recently synthesized RNA-dependent RNA polymerase (replicase) ribozymes [105,106,107] are highly complex and quite long (longer than the maximal length allowed by Eigen’s paradox). This is not surprising, given that RNA polymerization is a highly complex catalytic process that includes the ligation of nucleotides and the binding of template and copy strands, as well as their separation at the end of the process. Since ligation does not require a long ribozyme sequence [108], it is template binding and daughter-strand separation that necessitate the help of more complex and longer RNA polymerase ribozymes. Such a ribozyme complex has a very low chance of assembling in a short time, even from the huge random RNA population that we may assume to be produced by abiotic reactions in places such as near hydrothermal vents at the bottom of the prebiotic ocean [109]. However, considering the enormous amount of time at the disposal of prebiotic attempts to boot up RNA replication using a huge initial RNA pool with a fast turnover of random sequences, the assumption that some slow and vague ribozyme-aided RNA replication mechanism appeared by chance at some point and took the first evolutionary steps towards life seems not to be entirely remote. Such a self-replicating ribozyme replicase complex has not yet been discovered experimentally but neither have we spent eons of time looking for it in a practically infinite sequence pool. Wu and Higgs [110] suggest a simple model for a self-inducing evolutionary mechanism capable of producing replicase ribozymes with relatively high catalytic activity, starting from a very inefficient “primordial” replicase. 

With the ribozyme-based replicase function in place, another necessary condition of self-replication must be met: a continuous supply of activated nucleotide monomers in spite of the rapidly increasing monomer consumption by the exponentially growing RNA replicase population. Given that abiotic monomer production must have been very slow (if it occurred at all) without enzymes under prebiotic circumstances [109,111], we must assume that metabolism was also catalysed and the necessary catalytic help for monomer production came from within the random pool of RNA sequences as well. Any sequence that happened to have some catalytic activity capable of speeding up, at least to some extent, a metabolic reaction of the actual reaction network producing the monomers offered an indirect selective edge to the replicase, which, therefore, “adopted” the new sequence by giving it a replication advantage in exchange for the metabolic one received. Keeping the metabolic machinery running requires all the ribozymes of the system that contribute to the metabolic reaction network with their enzymatic activities to remain robustly coexistent. This is not an easy task for different species of replicators competing for the same limiting resource (the monomer pool) that they depend on for their replication. The ecological principle of competitive exclusion (the Gause-principle, see Section 2.1) permits the survival of just a single replicator species on a single limiting resource [15] but a single ribozyme cannot, obviously, catalyse all the chemical reactions of even a simple metabolic network. The mutual dependence of each metabolically indispensable replicator species on the presence of all the others may seem to alleviate the exclusion principle but it is easy to show that even the mandatory cooperation of the replicators is not sufficient for that to happen if the system is well-mixed without any local inhomogeneity permitted. With complete spatial homogeneity (and/or global mass interaction of the replicators and the metabolites they use and produce) assumed, the replicator system follows the simple mean-field dynamics (18)$${\dot{x}}_{i}=r_{i}x_{i}M\left(x\right)-\Phi_{i}\left(x\right)$$where **x** = (*x*_1_, *x*_2_, …, *x_s_*) is the population density vector for the *s* different, metabolically essential replicator species with replication rate vector **r**; *M*(**x**) is the monomer supply provided by metabolism at ribozyme densities **x**; and ϕ is an outflow vector ensuring that the total density $\overset{s}{\underset{i=1}{\sum }}x_{i}$ of all the essential ribozymes remains constant. In accordance with the assumption regarding the metabolic role of essential ribozymes, the metabolic function *M*(**x**) must take the value 0 if any one of $x_{i}$ is zero. A simple realization of this constraint is using a metabolic function proportional to the geometric mean of replicator densities:(19)$$M\left(x\right)=c{\left(\overset{s}{\underset{i=1}{\prod }}x_{i}\right)}^{\frac{1}{s}}$$where *c* is a positive constant. Since the metabolic effect of the actual monomer supply is the same for all replicator species at any particular moment, their relative (instantaneous) growth rates are determined by their (constant) *r_i_* growth parameters alone, i.e., the metabolic function has no effect on the order of the growth rates *r_i_M*(**x**) at any time. Therefore, the replicator with the highest growth parameter excludes all the others, in agreement with the Gause-principle.

The dynamics of the system are radically different, however, if the highly unrealistic assumption of complete global homogeneity postulated in the mean-field model is relaxed. The growing family of the Metabolically Coupled Replicator System (MCRS) models offer a chemically and ecologically feasible spatial mechanism for the robust maintenance of a metabolically active set of different ribozymes attached to mineral surfaces, assuming thatthe replicase function is given: any RNA sequence is capable of producing a copy of itself by template replication if it has a sufficient concentration of monomers at its disposal.all the members of the metabolic replicator set are indispensable in running a simple metabolic reaction network (metabolism) producing the monomers; if any one replicator type is missing from the set, monomers are not produced at all and the system goes extinct.the replicators are attached to a 2D mineral surface on which their horizontal mobility is limited; replicators leaving the surface are lost to the system (replicator “death”).nutrient compounds (external initial substrates of the metabolic reaction network) are supplied from the third spatial dimension in excess.the metabolites (substrates and products of the reactions that the replicators catalyse) are also attached to the surface, on which they may diffuse to a certain distance *d* before either being detached from the surface and lost, or used in a metabolic reaction or in replication (Figure 8).the metabolic contribution to the probability of a certain replicator being copied is dependent on the local ribozyme composition of its metabolic neighbourhood (i.e., within the distance *d* defining the metabolic neighbourhood of the focal replicator); only metabolically complete neighbourhoods (which have at least one copy of each essential ribozyme) allow for replication.the metabolically active set of ribozyme replicators may have enzymatically inactive parasites, i.e., replicators which do not contribute to monomer production but use the monomers produced by the cooperating replicators for their own replication.

##### Replicator Diversity and Ecological Stability in MCRS Models

Unlike the mean-field version, the stochastic (lattice) implementation of the spatially explicit MCRS model keeps all the metabolic replicators coexistent and shows robust ecological stability (Figure 9) [33,36,37]. Limited mobility and localized interactions of the replicators and the metabolites allow local group selection to operate: parts of the community lacking any one of the metabolic replicators are doomed to local extinction, pre-empting the habitat for invasion from nearby, metabolically complete neighbourhoods. The system is also resistant to its parasites in the sense that, even though parasitic replicators can invade the metabolically cooperating ribozyme community, they cannot destroy the cooperation altogether, because the damage they inflict on the system remains local and ephemeral. Wherever parasites take over locally, they stop monomer production and thus, indirectly, they commit suicide by disrupting their own monomer supply and starving themselves to death. This result is in line with that of Szostak et al. [112], whose model predicts parasite invasion in a replicase ribozyme population but without the parasites destroying the system. The coexistence of a replicase ribozyme and its parasitic quasispecies in a multilevel selection regime has also been demonstrated in [113].

For more than a decade of its development, the MCRS model family has been proven to be ecologically robust against many different changes in its basic assumptions and structure. Introducing trade-off relationships between replicator traits such as replicability and ribozyme activity [38], assuming variable system sizes and metabolic neighbourhood sizes [36], allowing ribozyme promiscuity (i.e., parallel or alternative enzyme activities of the same replicator) [35], the explicit assignment of the replicase function to an additional replicator species [34] or allowing for phenotype-genotype separation in the complementary strands of the replicators [115] did not change the general conclusion regarding the viability and resilience of the system.

Open chaotic flows (see Section 3.2.2) can offer an ideal cradle for MCRS as well. Stretching and folding moves different replicators close to each other along parallel filaments. Károlyi et al. [32] studied MCRS in open chaotic flows and they found that metabolically coupled replicators can indeed coexist in such habitats and they are also robustly resistant against their parasites (Figure 10).

Diffusion is omitted from the presented models. Since one has to consider only molecular diffusion in case of chaotic advection, this simplification is adequate. Molecular diffusion washes away fine fractal structure only below a critical length scale, while dynamical equations don’t change qualitatively [116]. Since particles move chaotically only for a finite time in open chaotic flows, replicators will not be washed out of the chaotic fractal if the time scale of replication is shorter than the time scale of moving along the fractal set. However, this condition is easily satisfied for example in the wakes of oceanic islands where particles may be trapped for months or even for years [117]. We have demonstrated that the coexistence of competing replicators is not a problem in open chaotic flows but evolvability is. Similar to parabolic replication, there is no selection in this habitat, so we have to assume other regions providing intense turbulent mixing where exponential dynamics still maintain selection [89]. That is, some problems of the formation of early replicator communities can be alleviated in open chaotic flows but this habitat alone is not a nostrum for all challenges. On the empirical side: open chaotic flows frequently occur in oceans, for example around islands or in hydrothermal vents. Recently [118] have shown numerically and experimentally that chaotic advection indeed accelerates surface reaction kinetics in the porous mineral substrates characteristic of sites near hydrothermal vents.

Besides the problem of maintaining sufficient amounts of information the other main challenge prebiotic systems had to face was maintaining the critical concentrations needed for reactions to occur at sufficient speeds. Particles are accumulated along the fractal set in open chaotic flows, so the physics of such habitats effectively solves the problem. We emphasize here that no fine tuning of the model is needed for this effect: an open flow is chaotic within a wide range of flow speeds. Changing the speed or direction of the flow doesn’t modify its main physical character.

##### Evolutionary Stability and Evolvability in MCRS Models

**Phenetic mutations.** Enzymatic control of RNA strand separation during the replication process guarantees that the metabolic replicator community does not follow parabolic growth kinetics (cf. Section 3.1.2) in MCRS, i.e., the populations of all replicator species have the capacity to increase exponentially and this is prerequisite for their evolvability (or, more precisely, selectability). Recent versions of MCRS allow for mutational changes in the structures of all metabolically active replicators so that evolutionary shifts in replicator traits can be simulated and their effects on coexistence and on the stability of the system as a whole can be analysed. In earlier models, only phenotypic changes in the most important replicator functions—replicability, rate of decay and metabolic (ribozyme) activity—had been considered. The phenetic models [35] defined reasonable yet arbitrary trade-off relationships among the three critical traits, assuming, for example, that a mutant replicator that is easier to copy (i.e., features a higher replicability) than its template is less likely to be as good a catalyst (i.e., it has weaker metabolic enzyme activity) and it is probably more exposed to environmental effects, leading to faster decay or loss from the surface (i.e., its decay rate is higher)—all for the very same and, in these phenetic models still implicit, structural reason: a less compact, looser secondary structure. It can be shown in simulations that—even with rather strict phenotypic trade-off constraints enforced between different aspects of replicator performance—it is possible to evolve metabolic replicator sets of nearly optimal values in all these three traits if a small “wobbling” is permitted in the trade-off relationships [35].

**Genetic mutations.** The necessary level of wobbling may indeed be provided by the thermodynamics of RNA folding, as it has been demonstrated in the latest versions of MCRS [38], in which the purely phenotypic, sequence-implicit approach has been relaxed with the assignment of actual nucleotide sequences and the corresponding secondary structures (the latter calculated by the ViennaRNA algorithm [119] on the basis of free energy minimization) to all the replicators present or appearing by mutation in the system (Figure 11). The three critical traits of each sequence on the lattice are direct explicit functions of its primary and secondary structural (i.e., sequence and folding) features. The MCRS mechanism imposes selection on the variations of the resulting phenotypes. The most surprising feature of the dynamics of the extended system is its extreme propensity for robust replicator coexistence through evolving different metabolic functionalities (i.e., distinct ribozyme activity patterns) embodied in replicators of different sequences but highly similar population dynamic and enzyme kinetic properties. For example, simulating the sequence-explicit MCRS with three necessary metabolic ribozyme activities (blue, red and green in Figure 12) and a potentially infinite pool of different parasitic sequences (grey colour in Figure 12), starting from a random sequence distribution with different initial replicator lengths, converges to a stationary distribution with highly similar densities, lengths and enzymatic activities in the evolved set of distinct metabolic replicator species (Figure 12).

Previous purely ecological (i.e., non-evolving) MCRS models [36] have shown that the metabolically coupled spatial replicator system is robustly coexistent, even if the dynamically relevant parameters of the different replicator species are fixed at quite different values. The new sequence-explicit, evolving MCRS model automatically adds another layer of robustness to the dynamics by converging the dynamically and functionally important traits of the metabolic replicator species to quite similar values, which, of course, makes it easier to keep them coexistent. The ensuing almost-even density and activity distribution of the metabolic replicators is also advantageous for the efficiency of metabolism, which runs best in metabolic neighbourhoods consisting of equal copy numbers of the different ribozymes, since the metabolic function *M*(**x**) is proportional to the geometric mean of the copy numbers. Obviously, mutations produce metabolically useless parasites in large numbers but they are quickly eliminated by the local regulation mechanism explained in the previous section.

The ecological stability and the parasite resistance of the MCRS is spectacular at a substantial range of its parameter space but the number *s* of metabolically essential replicator species sustainable by its simple diffuse group selection mechanism is always limited [36]. That is, if the first steps towards life had been taken as assumed in the MCRS scenario, the “chromosomization” of the RNA molecules and the separation of genetic and enzymatic functions must have had occurred at a relatively early stage of replicator evolution, because the number of ribozymes necessary to catalyse an ever more complex metabolic reaction network is well above the few that the early types of MCRS could have kept coexistent to form a stable replicator community. Chromosomization and genetic/phenetic role separation into complementary RNA strands are already being studied using other models compatible with the MCRS concept [27,120,121,122]; development of future MCRS models will also take that direction.

## 4. Discussion

The origin of life on Earth is one of the ancient enigmatic questions that humankind has always been asking. Besides the multitude of metaphysical and philosophical answers provided by different forms of civilization in our history we have not, until quite recently, seriously attempted to answer the question of the origin making use of the scientific methodology. There are a few strong reasons for this conspicuous delay in the scientific response to such a fundamental and ancient challenge. Life as we know it is a unique phenomenon, confined to our planet according to our present knowledge. We have no “independent experiments” pertaining to the origins of different forms of life from different points of the Universe at our disposal for comparison. For essentially the same reason it seems impossible to come up with a proper definition of what life is: any such definition attempt suffers from being built upon “ad hoc” assumptions. Yet another reason is the complete lack of fossil record that could channel our speculations on what actually happened almost four billion years ago in the prebiotic ocean. All we can count on is our conviction that the laws of physics and chemistry are time invariant and, therefore, we may be able to invent prebiotic scenarios that are in agreement with those eternal laws and thus, hopefully, their feasibility can be verified or falsified experimentally in the laboratory. This conviction governs our search for prebiotic system models satisfying the three conditions of diversity maintenance, ecological stability and evolutionary stability, which are feasible to apply to each model candidate in the following order.

The main criterion that has to be met is the ability of the model to maintain diversity. A system compliant with the diversity criterion can be ecologically stable or unstable. Provided that the model is ecologically stable, the next relevant question pertains to its evolutionary stability. An ecologically stable system that is also evolvable and stable against its characteristic parasites, i.e., one that meets all three criteria, may be a hopeful candidate for representing a possible prebiotic scenario. With at least one pillar missing (including evolvability, which is a prerequisite of evolutionary stability), the model cannot be considered as the basis of a realistic scenario.

The most promising such prebiotic evolutionary scenario is that of the RNA World [48,98,123], which has many different incarnations as regards their assumptions of the actual physical-chemical habitats of the ancient RNA populations, as well as the abstract structures of the dynamical models these assumptions imply. The models we have analysed and compared are directly relevant to prebiotic replicator dynamics with explicit or implicit reference to the RNA World but many of them have obvious relevance at higher levels of recent biological organization as well.

Even the simplest of chemical systems capable of evolutionary change must have featured exponential population growth—a capacity that is inevitably constrained in a finite world sooner or later. Out of a number of different competing entities, all capable of exponential growth, only a subset of the entities will survive due to the effect of competitive exclusion. If the entities compete for a single resource (or, in general, a single regulating factor), then there can be just a single survivor type and thus diversity cannot be maintained. This is the basic problem of prebiotic evolution (and, in fact, ecology and evolution on any level of organization) that has to be solved for a diverse system to be viable. This condition is met one way or another in all the models studied.

There is no definite answer to the question of what kind of diversity (and the corresponding quantity of information) would have been necessary to be maintained in a replicator system for it to be able to operate a prebiotic system of sufficient complexity. What we can do is to estimate the genome size of what is called a “minimal cell,” on the basis of a top-down analysis [124] but the result is still in the order of hundreds of genes. Of course, this huge information content is sufficient to operate the entire core of the machinery of recent bacterial life, which is certainly much more complex than what might have been the starting point of chemical evolution. With no available clues about the beginning in recent forms of life we are forced rely on models of simple replicator interactions in finding feasible prebiotic scenarios. The problem has been first addressed in Eigen’s quasispecies model (see Section 3.1.1) and the first quantitative solution offered to the diversity problem therein was the hypercycle (HC), the ability of which to maintain diversity may be considerable (apart from its dynamical stability issues). The same diversity maintaining capacity is inherited by the spatial (SHC) and the vesicular (CHC) versions of the hypercycle. The ability to preserve replicator diversity is limitless in both the parabolic replicator (PR) and the open chaotic flow (OCF) models. Even though these two models may seem dissimilar to the extreme in their assumptions (complementary strands replicate in an unstructured environment in one and autocatalytic replicators in an open chaotic flow in the other), they yield the same dynamics for essentially the same reason: the sub-linear dependence of replicator growth rates on replicator densities and the ensuing general advantage of rarity for all replicator types. The toy-versions of the MCRS model have a capacity to maintain diversity at about the order of ten different replicators [36] but the chemically more explicit—and dynamically more stable—versions have not yet been studied from this aspect. The effect of mitigating competitive exclusion has been verified in the stochastic corrector model (SCM) for up to a hundred different replicators [30]. Quickly replicating parasites and high mutational rate can still put a more stringent limit on the maintenance of diversity in the SCM.

Obviously, the results summarized above reflect the present state of the art for all model types discussed above, one of whose common assumptions is the omission of explicit replication chemistry: they consider the production of the daughter strand of a string replicator as a single-step reaction, disregarding the dynamics of nucleotide insertions. Implementing more detailed dynamics (considering changes in nucleotide supply, possibly for each monomer species, or the production of unfinished strands; etc.) may have a profound effect on the ability of each model type which in their implicit versions can maintain unlimited (PR and OCF) or nearly unlimited (HC, SHC and CHC) diversity. Such studies are yet to be carried out. We note here that the sequence explicit version of the MCRS does not lose any of its capacity to maintain diversity compared to that of the toy versions; on the contrary: it has a good chance to have it increased but this has yet to be tested.

In accordance with the intimate cross-dependences among the three dynamical features that we consider as the main criteria for evaluating models of prebiotic evolution, each model must be scrutinized from the viewpoint of its potential to preserve diversity in the face of the environmental changes characteristic of the supposed habitat of the replicators. In this context, we must consider environmental changes affecting the growth rates of the replicators (for example, in the form of additive mortality). Lacking individual boundaries and homeostatic regulation, resistance to such external effects must have been of profound dynamical importance in prebiotic replicator systems. Even if the “ecological” and “evolutionary” timescales were convoluted at the time, it must have been a necessary condition for any such system to be persistent that the composition of its species pool was dynamically stable. The hypercycle (HC) model is an underperformer in this respect: hypercyclically connected loops of over five species in size show wide fluctuations in replicator density even in response to small environmental perturbations. This means a high risk for one of the actual low-density members of the cycle to shrink below a critical level and disappear and—due to the collective autocatalytic coupling—for the whole system to collapse as a consequence. External disturbances destroy the mesoscale spatial symmetry of the spatial hypercycle (SHC) model but the corollaries with respect to the diversity of the system have not yet been studied. The SCM has not been analysed rigorously in this regard either but we can guess that random disturbances cannot have a strong deleterious effect on a system that is kept alive by random assortment in the first place. The wrapped hypercycle (CHC) model has not been investigated for disturbance resistance either but we expect it to inherit the probable weak response of the SCM. The disturbance responses of the parabolic replicator (PR) model and the open chaotic flow (OCF) model are in all probability similar to each other because of the dynamical homology of these systems but a formal analysis of the models in this respect is yet to be accomplished. The MCRS models have shown robust ecological stability against changing the replicator degradation rates (which correspond to environmental perturbations), by applying both sequence-length-dependent and -independent decay rates [33,36].

An obvious prior condition for a system to be evolutionarily stable is that it is evolvable. This criterion is not met by the parabolic replicator (PR) and the open chaotic flow (OCF) models in their present form, for the same dynamical reason that makes them capable of maintaining any level of diversity. It is, however, worth mentioning here that these approaches can be extended in ways allowing for Darwinian selection to operate on them for at least some of the time, thus potentially rendering them evolvable. Lacking detailed studies of this problem we cannot claim more in this regard at the moment. The hypercycle (HC) model has severely limited evolvability: first it is highly improbable to have mutation events that lead to an increase in the number of dynamically coupled members in a hypercyclic loop; more importantly, such a new hypercycle cannot increase in frequency, because the hyperbolic growth law governing their dynamics favours the old system which has a higher initial density. Therefore, the capacity of HC to increase the information content it replicates is weak. In addition, it is highly sensitive to the occurrence of parasitic mutants: both selfish and shortcut parasites may destroy the autocatalytic loop. Most of these problems occur in the spatial version (SHC) as well, except for the sensitivity to selfish parasites which it resists; the evolvability and the shortcut parasite resistance of the SHC model are just as bad as those of the non-spatial (HC) version. The stochastic corrector model (SCM) is evolvable and evolutionarily stable, with evolution acting on two levels: among replicators within the same compartment and among the compartments of different replicator composition. (Note that the corresponding group selection mechanism acts on higher organizational levels as well, to which the SCM may, therefore, be also applicable. At these higher levels, the compartment boundaries are usually naturally given, unlike in prebiotic SCM systems which assume the compartments being present and capable of coordinated fission, without explaining their origin.) The compartmentalized hypercycle (CHC) model is resistant to both kinds of its potential parasites due to the stochastic correction effect of group-level selection but it is just as limited in its evolvability as the HC and SHC models are. The MCRS model meets all the conditions of evolvability and parasite resistance: it can adopt (a limited number of) new metabolic replicator species as long as they contribute to a more efficient metabolism. The new metabolic replicators may originate as mutants of parasitic sequences, which comprise quasispecies around the existing metabolic ribozymes but cannot destroy the system, thanks to the parasite control through metabolic efficiency within the diffuse group structure of metabolic neighbourhoods. Notice that the group selection mechanism works in the MCRS without assuming compartments of unexplained origin and, in fact, it also offers a plausible (but so far not implemented) scenario for the evolution of membrane-producing ribozymes by parasite adoption. Table 2 summarizes these results.

In summary, the present state of the art of prebiotic string-replicator models suggests that the three most promising directions for modelling prebiotic ecology and evolution seem to be:the stochastic corrector model (SCM), as long as the origin and the maintenance of compartments coupled to replicator population growth are explained;the parabolic replicator (PR) model and its dynamical homologue, the open chaotic flow (OCF) model, with the future addition of a scenario for the appearance of a selection regime; andthe metabolically coupled replicator system (MCRS) model, which meets all the criteria for maintaining diversity and being robust both in the ecological and the evolutionary sense but only for a limited number of replicators as yet.

In our opinion, the MCRS scenario seems to be the one that is built on the most plausible set of assumptions and offers the best perspectives for further research on replicator evolvability. However, a few words of caution are due at the end of this survey. Even if we find a plausible, sufficiently detailed, ecologically and evolutionarily stable scenario for the origin of life, proving that chemical evolution had followed a blueprint resembling—at least in the most important respects—that scenario in creating life from non-life 3.8 billion years ago seems almost impossible, mainly because we lack fossil evidence. Even the laboratory verification of the feasibility of any specific scenario is a remote possibility for now, given that our present models considering chemical details such as the kinetics and thermodynamics of the reactions involved are still incapable of being empirically instructive. What we currently have at hand is but a stepping stone to future research aimed at the distant target of once re-creating life in the lab.

## Figures and Tables

**Figure 1 life-07-00048-f001:**
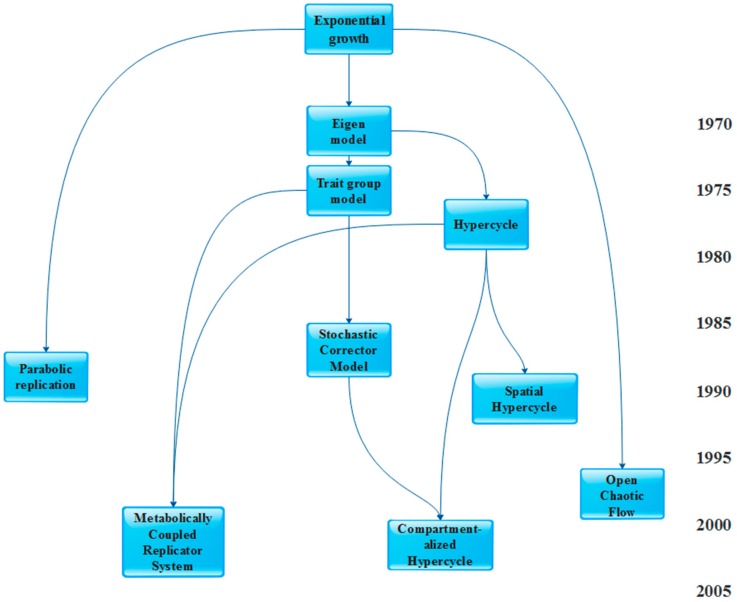
Genealogy of prebiotic replicator models. The simplest possible model for replicator dynamics is exponential growth, which does not allow coexistence as the fittest always wins. Since it is an idealistic case, all sorts of extensions and deviations from the basic model are intended to make prebiotic systems more realistic and more permissive in terms of coexistence, ultimately crossing the barrier beyond which a sufficient amount of information can be stably maintained on the evolutionary timescale for cellular life to emerge.

**Figure 2 life-07-00048-f002:**
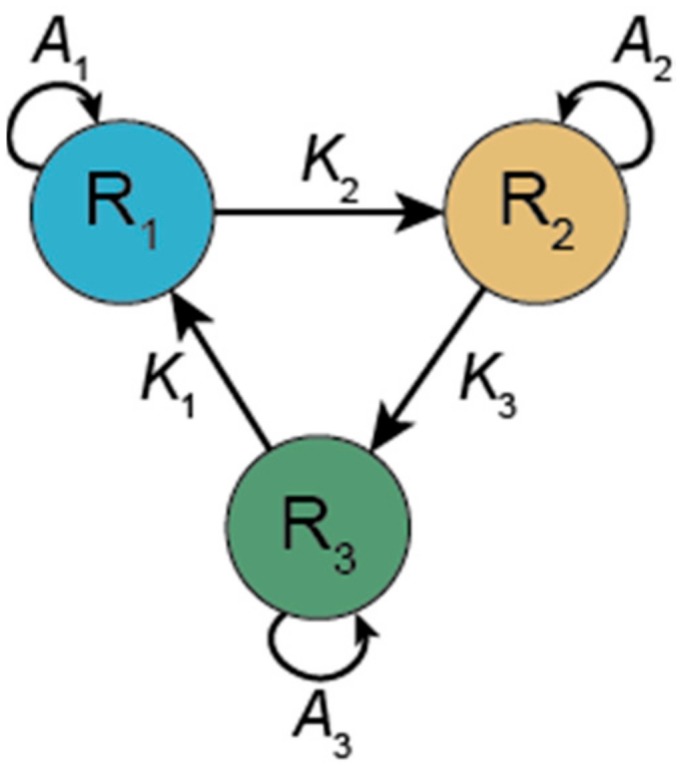
A 3-membered hypercycle. Each member (R*_i_*) of the hypercycle can catalyse its own replication (A*_i_*) and the replication of the next member in the cycle (*K_i_*_+1_).

**Figure 3 life-07-00048-f003:**
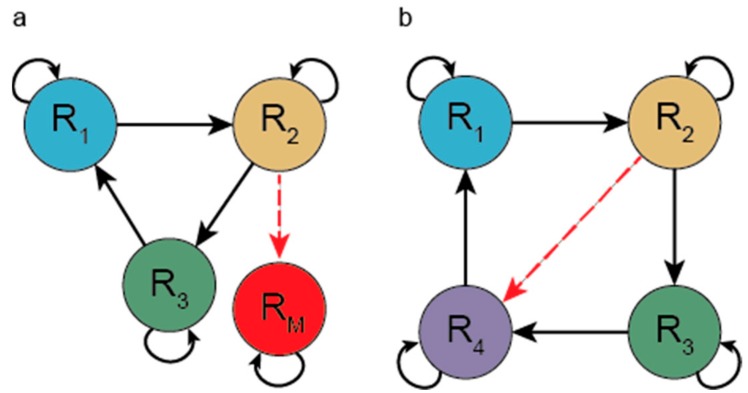
Evolutionary instability in the hypercycle. (**a**) A parasite (R_M_) that enjoys catalysis from a member of the hypercycle (R_2_) but does not take part in the hypercycle organization. (**b**) A shortcut mutation (red dotted arrow) which changes the specificity of the catalysis offered by a member of the hypercycle (R_2_) so that it catalyses the replication of a member it should not catalyse. R_1_, R_2_ and R_4_ now form a 3-membered hypercycle, which can replicate faster than the 4-membered hypercycle.

**Figure 4 life-07-00048-f004:**
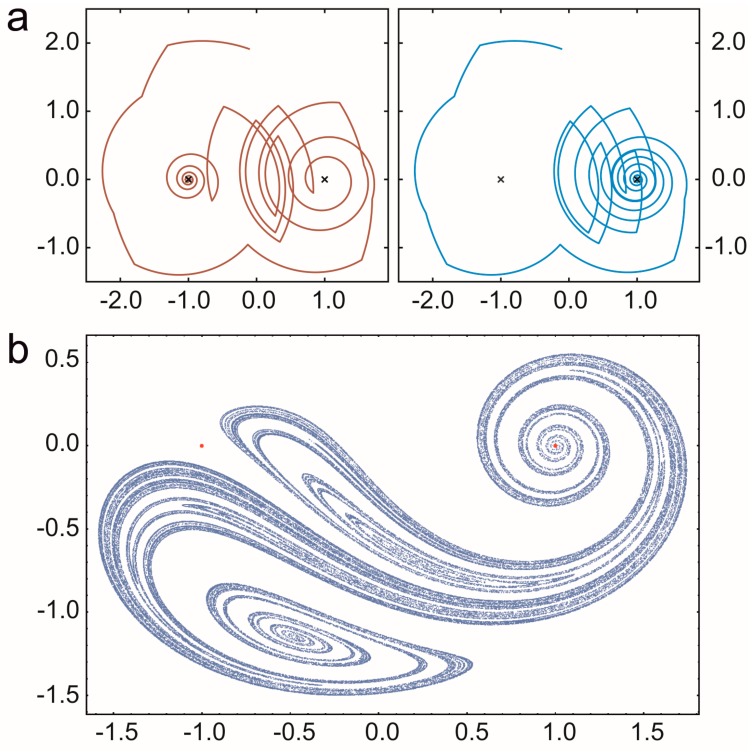
The motion of particles in an open chaotic flow. The blinking vortex-sink system is used for demonstration. It models the outflow from a large bath tub with two sinks that are opened in an alternating manner. Crosses denote the sinks. (**a**) Diverging trajectories of two particles that initially are close to each other. In this example, they even leave the bath in different sinks. (**b**) A snapshot on particles distributed along a fractal set (chaotic saddle) in open chaotic flow generated by the blinking vortex-sink system. (Based on [32].)

**Figure 5 life-07-00048-f005:**
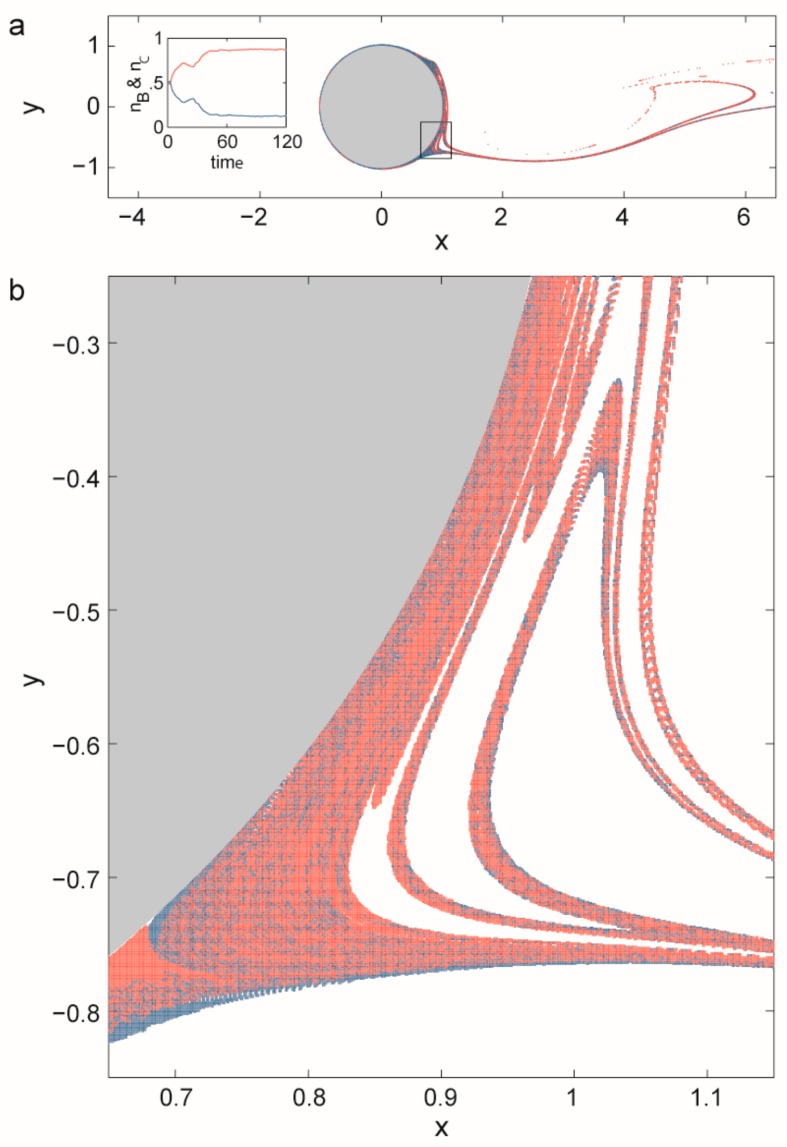
The distribution of two replicators (red) B and (blue) C competing for the same resource material (white) in the wake of a cylinder. The flow is from left to right. The inset in (**a**) shows the time-dependence of the population numbers *n*_B_, *n*_C_ and clearly indicates the approach to a steady state of coexistence. A blow-up of the region indicated by a rectangle in (**a**) is seen in (**b**). B-s replication rate is 4/3 of the C-s, while decay rates are the same for the two species. Coexistence of 35 species is experienced in other simulations. (Based on [31].)

**Figure 6 life-07-00048-f006:**
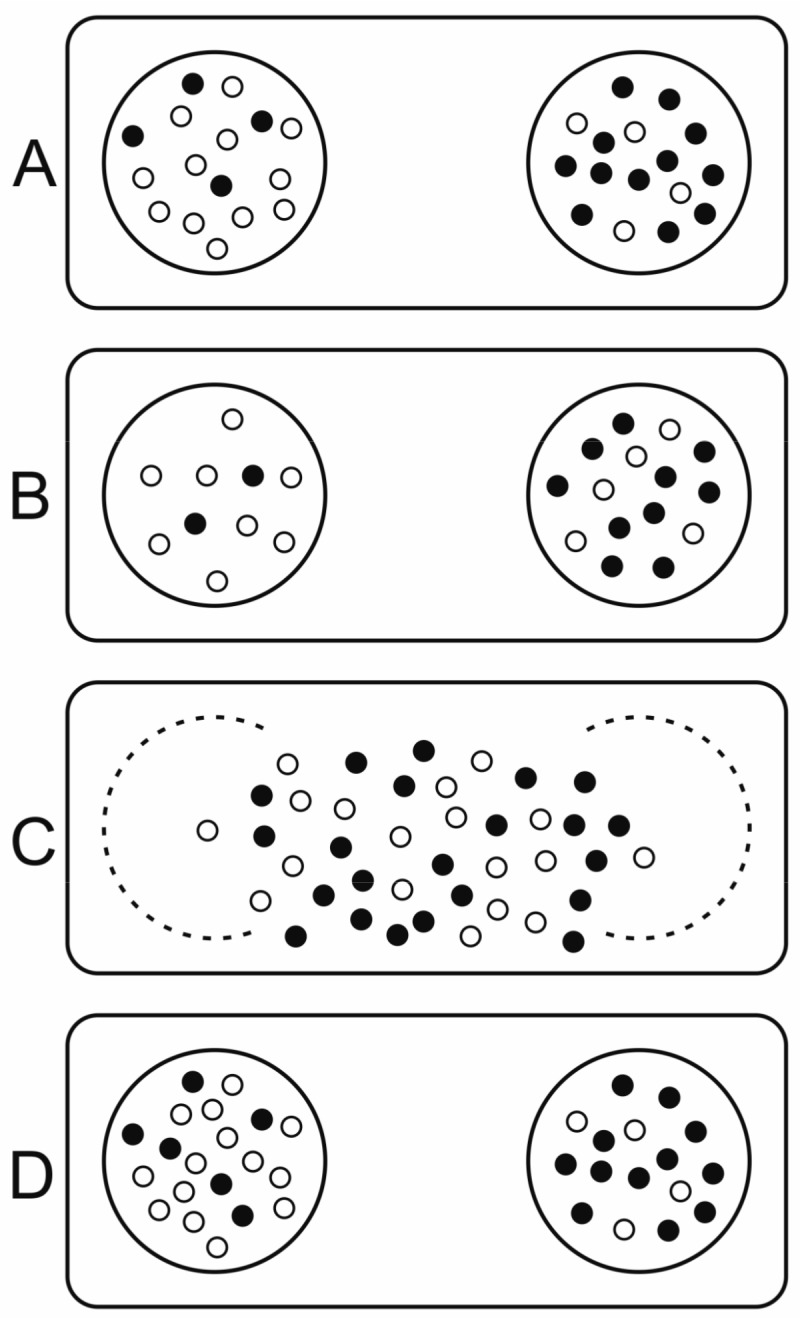
Wilson’s trait group (or structured deme) model. (**A**) Individuals with different traits, black and white dots, form localized trait groups. (**B**) After ecological interactions and selection, (**C**) survivors are released to form a pool, where they can reproduce. (**D**) New groups form (After [39]).

**Figure 7 life-07-00048-f007:**
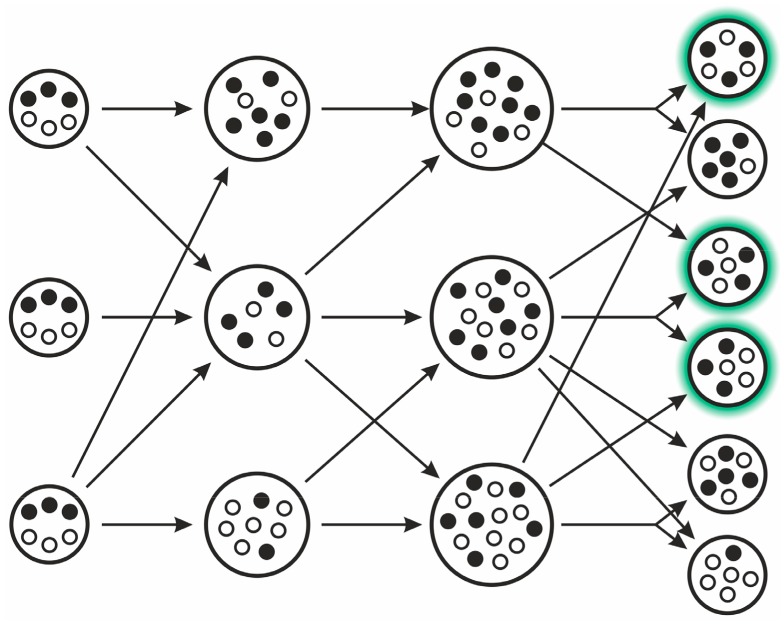
The stochastic corrector model. The two replicator types are indicated with filled and empty small circles. Arrows indicate transitions, as individual compartments grow and divide. Compartments with green highlight (after division) contain the optimal composition of replicators (after [29]).

**Figure 8 life-07-00048-f008:**
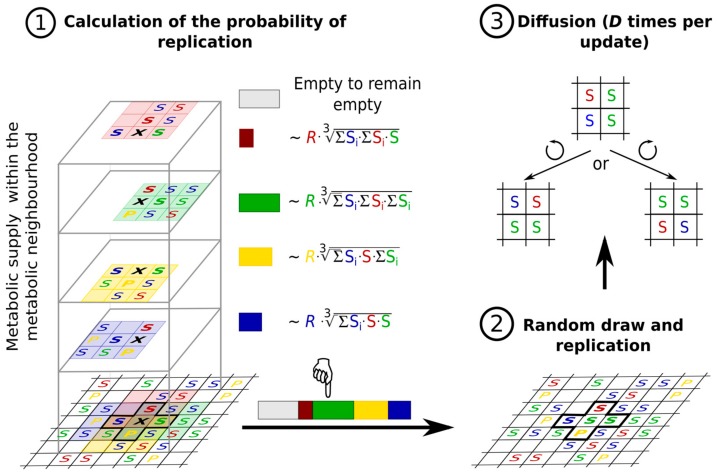
Basics of the MCRS algorithm. (1) Metabolic support of the four replicators in the von Neumann neighbourhood of an empty site (black **X**). Red, green and blue **S**s denote different, metabolically active replicator species, the yellow **P** stands for a parasitic replicator. (2) The replicator taking the empty site by the next generation is determined by a random draw, with the empty site to remain empty having a constant claim and the claim of each adjacent replicator depending on its own replicability (*R*) and the metabolic support it receives from within its own 3 × 3 metabolic neighbourhood. (3) Each replication step is followed by replicator diffusion, implemented as *D* elementary steps of the Toffoli-Margolus algorithm [114] at random positions of the lattice.

**Figure 9 life-07-00048-f009:**
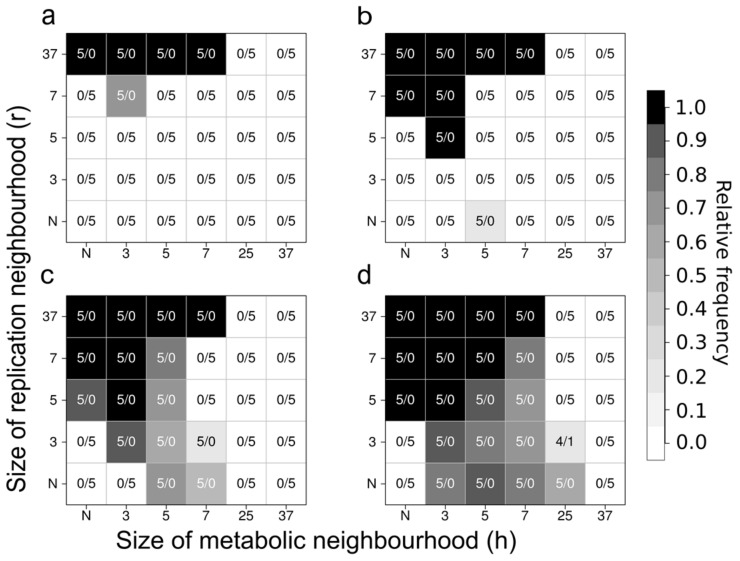
Persistence of the MCRS at different sizes of the metabolic and the replication neighbourhood. Neighbourhood sizes are given as side lengths of a square-shaped section of the lattice that is centred on the focal site; *N* stands for von Neumann neighbourhood. The *i*/*j* values inside the table cells specify the numbers of persistent/extinct systems out of five replicate simulations; grayscale values are system densities in percentages of sites occupied by replicators within the lattice after 10,000 generations. Panel (**a**) *D* = 0, Panel (**b**) *D* = 1, Panel (**c**) *D* = 4 and Panel (**d**) D = 100 (Based on [36]).

**Figure 10 life-07-00048-f010:**
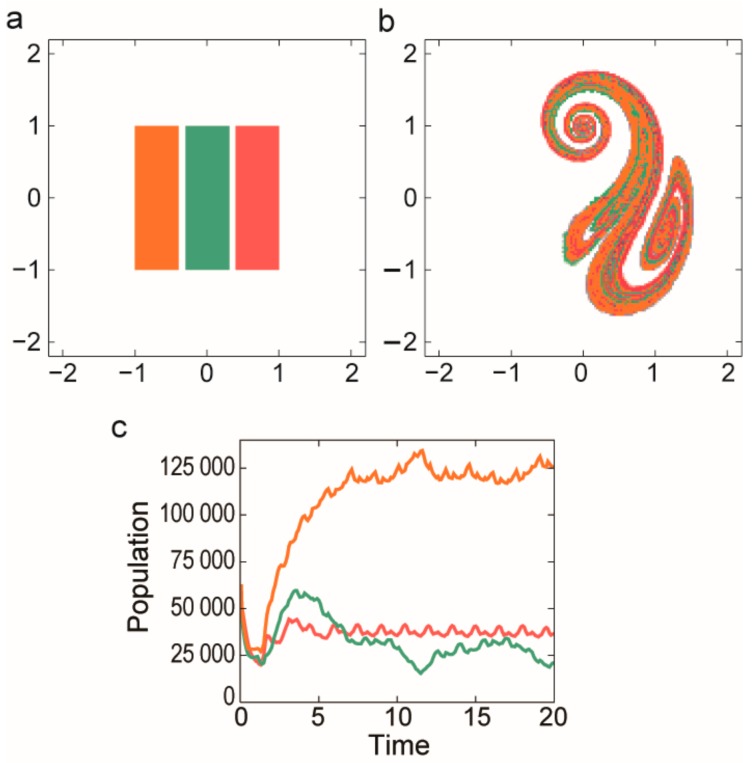
Initial distribution of the replicators a snapshot and time dynamics of the metabolic network on chaotic advection by an open flow. (**a**) Replicators are placed into separate stripes initially. Different species are denoted by different colours. (**b**) The snapshot of spatial distribution of replicators at *t* = 10 in units of flow’s period. (**c**) The population size is shown as a function of time measured in units of the flow’s period. The size of the metabolic neighbourhood was σ = 10 for each competitors and their spontaneous decay was δ = 0.02. The replication constants were different for each species, these were *k*_1_ = 3 (red), *k*_2_ = 4 (green) and *k*_3_ = 5 (orange). The potential that an empty site remains empty was *C*_e_ = 2 (Based on [32]).

**Figure 11 life-07-00048-f011:**
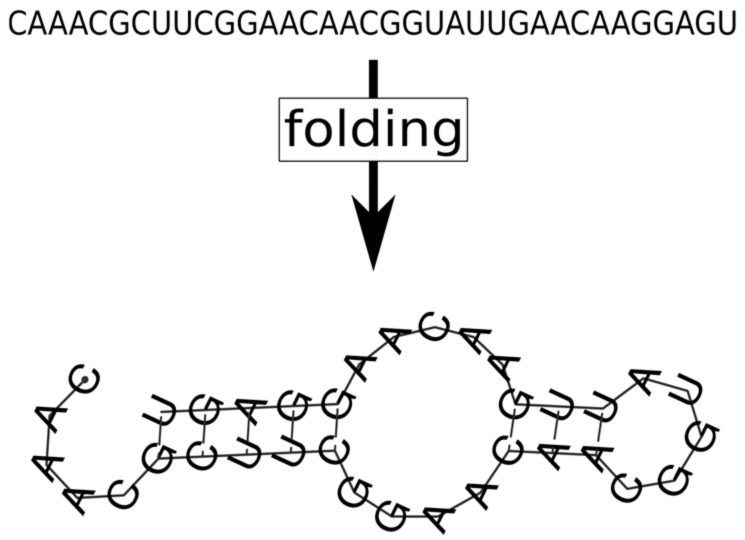
The 2D secondary RNA structure is determined from the primary structure (nucleotide sequence) using the thermodynamic condition that the folded molecule should have the smallest possible free conformation energy. The conformation calculations are executed by the ViennaRNA algorithm.

**Figure 12 life-07-00048-f012:**
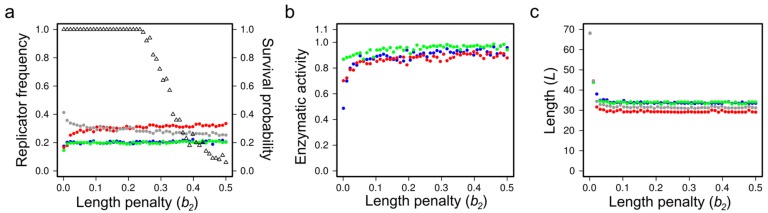
Trait convergence in the sequence-explicit version of MCRS. (**a**) Relative replicator frequencies, (**b**) (metabolic) ribozyme activities and (**c**) replicator lengths at the stationary states of the simulations, after two million generations, as functions of the selection pressure against sequence length (“length penalty”). Open triangles in panel (**a**) are the proportion of surviving systems out of 100 parallel simulations; Red, green and blue dots represent the three different metabolically active replicator types, grey dots represent all the parasitic (metabolically inactive) replicators. Persistent MCRS systems are efficiently selected for convergence in all the fitness-related traits of the replicators (Based on [38]).

**Table 1 life-07-00048-t001:** Categorization of dynamical models with respect to their temporal and structural resolution. For details of the models see the main text and references. Note that unstructured replicator models in discrete time are generally lacking as fully (i.e., in both space and time) continuous models are much easier to handle analytically.

Structure/Time		Discrete Time	Continuous Time
Without structure (only global interactions)		-	QS, HC, PR
With structure (global and local interactions)	*Compartmentalized*	SCM	CHC, TGM
	*Spatial*	MCRS	SHC, CM

**Table 2 life-07-00048-t002:** An assessment of each model in the context of the three main criteria and their “evolvability”, the scope for the adoption of mutant replicators with a useful function into the system.

	Diversity-Maintaining Ability	Ecological Stability	Evolutionary Stability	Evolvability
**HC**	An arbitrary number of sequences can coexist if there is no population stochasticity; otherwise some species can be lost.	The cooperative nature of organization ensures that, given high enough catalytic aid, the system is stable.	Selfish parasites and short-cut mutants can destroy the system.	No
**SHC**	Due to the importance of local interaction, the number of potentially coexisting sequences is limited.	The cooperative nature of organization ensures that, given high enough catalytic aid, the system is stable.	The organization is stable against selfish parasites but short-cut mutants could still take over. The system still cannot evolve new hypercycle members.	No
**CHC**	The number of sequences is limited due to the random assortment into daughter cells.	The cooperative nature of organization ensures that, given high enough catalytic aid, the system is stable.	Group selection can probably maintain existing diversity but the system still cannot evolve new hypercycle members.	N/A
**PR**	An arbitrary number of sequences can coexist at arbitrarily small concentrations.	The continuous advantage of rarity of any replicator ensures coexistence at any external parameter combination.	- Non-Darwinian regime. - No classical selection. - Any number of new replicators can invade the community without outcompeting others.	No
**SCM**	N/A	N/A	- Stochastic replication/degradation. - Random assortment during fission. - Shared metabolism.	Yes
**OCF**	An arbitrary number of sequences can coexist at arbitrarily small concentrations but locally dense populations (the concentration at the boundary of fractals can be very high).	The continuous advantage of rarity for any replicator ensures the coexistence at any external parameter combination.	- Non-Darwinian regime. - No classical selection. - Any number of new replicators can invade the community without outcompeting others.	No
**MCRS**	A limited number of ribozyme replicators coexist in a highly robust system.	Advantage of rarity due to the mandatory metabolic cooperation of all replicator species maintains stability across the parameter space.	- Darwinian selection for fitness homogeneity. - Dynamical trait convergence with functional diversification. - Sequence-dependent replicator functionality. - Parasite resistance. - Parasite “adoption” for useful functions. - No need for membrane compartments.	Yes
**TGM**	N/A	N/A	- Small compartment size. - Low diffusion rate. - Rapid extinction of inferior compartments. - Selfish replicators are coupled to cooperative ones.	Yes

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
