# Peer review of "Ecology and Evolution in the RNA World Dynamics and Stability of Prebiotic Replicator Systems"

_life, 2017, doi:10.3390/life7040048_

Round 1

Reviewer 1 Report

This review is thorough and should be valuable to others in this area of astrobiology. It may not be as useful to those who are not familiar with the specific area due to the highly specialized content. Nevertheless, the paper examines existing approaches to modeling an RNA World and through careful comparison, draws significant conclusions upon which to build future work of this kind. The only weakness I find is the highly restricted nature of this work, which seems so theoretical as to be isolated from considerations of the actual complexities of the real early Earth that will certainly have placed more constraints and, perhaps, opportunities than recognized by any of these models.

Author Response

We agree that the present state of the field is highly speculative, mainly due to the relative scarcity of relevant experiments. There are, however, well established empirical methods for testing the chemical feasibility of certain details of the models covered in the review. Obviously there may be relevant mechanisms not yet studied, and some of the mechanisms assumed in the models may prove irrelevant in the light of future experiments, but this should not stop us analyzing theoretical constructs which seem feasible. 

Reviewer 2 Report

This paper is very conceptual and hypothetical. I cannot evaluate this paper correctly. This paper may not suitable as review paper but suitable as hypothetical original paper.

Author Response

See our reply to Reviewer #1. We do not agree that reviewing the present state of the art in the research of prebiotic evolution would not be possible we hope to have proven otherwise.

Reviewer 3 Report

The paper is very well-written and presents the comprehensive review of models related to replicator systems that try to explain origins of life. Authors collected plenty of useful information in one review article and compared them using thee different criteria. 

My most important concern is related to the mixing of modeling methodology with models themselves. From the article, it is not clear what was the reason for such selection of reviewed model. For example, authors differentiate between Hypercycle and Spatial Hypercycle which are the same model but modeled using different methodologies (differential equations and cellular automaton, respectively). Authors should define, starting from the abstract, if they are compering models or implementation of models using some specific methodology.

Moreover, I have following minor remarks:

Line 108: There is also an interesting method for modeling space using multi-agent systems that have some advantages over cellular automata. It should be mentioned. For example, see: http://journals.plos.org/plosone/article?id=10.1371/journal.pone.0180827

154: “the takes” – some word is probably missing

174: I think that it should be mentioned that the concept of evolutionary stability has strong theoretical foundations in evolutionary game theory: book by Smith’82 Evolution and the Theory of Games

208: From the table, it follows that there are no discrete models without structures. However, there are such models, for example, based on non-spatial Gillespie method. For example see: https://www.ncbi.nlm.nih.gov/pubmed/19777150 If you do not want to review them you should at least explain why and list them.

208: Why do not you consider replicases-parasites model (http://journals.plos.org/plosone/article?id=10.1371/journal.pone.0180827, http://journals.plos.org/ploscompbiol/article?id=10.1371/journal.pcbi.1000542)? Again, it should be at least mentioned, especially that you describe an example based on it later (DIV virus in the section about TGM).

224: There is a recent article summarizing Eigen’s work on Hypercycle that should be referenced here: http://journals.plos.org/ploscompbiol/article?id=10.1371/journal.pcbi.1004853

268: I think it should be K_{i+1} not K_i in the figure’s description.

271: Formally, you should mention that according to your notation x_0 (for i = 1, x_{i-1} = x_0) is an alias for x_n. 

282: In my opinion, it is not clear from the description that for n > 4 hypercycles are unstable.

307: What is the meaning of A? It is not explained.

382: “replicator concentration” – which replicators: free, associated (AA) or total, counting each associated as two?

383: Could you define the relation between r_i and (K, k, k’)?

395: “(11)” – do you mean “Eq. (11)”?

405: You use a different style for >> operator than in line 374.

453: I think that the great achievement of Boerlijst and Hogeweg was the demonstration that spatial consideration could completely change the behavior of the system, for example solving the problem of global extinction for large systems. I think that you should add some comment on it in the introduction to section 3.2.

492: What happens on “reaction front within a fluid layer”? Please define the reaction similarly to lines 371 and 274.

514: Figures are not marked with (a) and (b) labels. You use it only in figures description.

524: Equation has a missing number.

569 and 591: “)” is missing.

600: Add comment what empty and filled circles mean.

625: As we cannot differentiate x and y, I propose to use x_1 and x_2 instead.

639 and 643: “)” is missing.

682: I think that “highly improbable” is too strong wording. We still do not know many mechanisms, and I believe that still there is a possibility for many groundbreaking discoveries. Let’s take CRISPR-Cas as an example of the relatively simple mechanisms that needed years to be understood. The same can still happen in the area of replication.

699: I do not agree that the time on prebiotic Earth was sufficient to construct working replicases by chance. For an interesting analysis of this chance in context of probabilistic resources existing in the universe see: https://www.cambridge.org/core/journals/european-review/article/understanding-life-a-bioinformatics-perspective/7A0FC37981D2DDE7F8B7D58C26F3D88F which can also be cited here. I am not a creationist, and I do not think that some external force had to influence the creation of replicases, but it is almost certain that some physical or chemical mechanism for supporting it had to exist.

728: “outflow vector” instead of “outflow term”.

766: “Replicator taking…” is probably easier to understand than “Which of the replicators take…”

Figure 9: What is the difference between A, B, C, D lattices?

800: Put the author name before [31].

Figure 10: No A, B, C labels above pictures.

836 and 854: I think it can be beneficial to bold two first words (mutation types). It will be easier to read.

854: Paragraph starting with “That” does not sound good.

861: Folding is tertiary structure. The secondary is topology (interactions between nucleotides)!

866-7: “in Figure” not “on Figure”.

Table 2: Description says about three criteria, but you have four columns.

I do not have any remarks related to the discussion. It is clearly written and summarizes described models correctly. I am sure that after improving the article according to the comments listed above, it is certainly worth publishing as an article giving excellent overview of research related to simulating origins of life. 

Round 2

Reviewer 3 Report

Thank you for the detailed explanation how you intended to organize the paper. I accept your arguments. The only remark that I have is that one more sentence of such explanation can also be added in the Introduction to make it more transparent for the reader.

Author Response

In accordance with your suggestion, we have extended the introduction.